# DRIFT: DATA REDUCTION VIA INFORMATIVE FEATURE TRANSFORMATION - GENERALIZATION BEGINS BEFORE DEEP LEARNING STARTS

## ABSTRACT

Despite the remarkable optimization power of modern deep neural networks, robust generalization remains critically dependent on the quality of input representations. High-dimensional pixel data is plagued by noise, redundancy, and spurious correlations that hinder stable learning and widen the train-test generalization gap. We introduce DRIFT (Data Reduction via Informative Feature Transformation), a lightweight, physics-informed preprocessing method that reinterprets images as static displacement fields of a thin elastic plate under simply supported boundary conditions. By projecting each image onto the analytically derived orthogonal basis of vibrational mode shapes, low-frequency sinusoidal patterns governed by the biharmonic equation, DRIFT yields compact, interpretable, and intrinsically smooth features that emphasize energetically dominant spatial deformations while suppressing high-frequency noise. Extensive experiments on MNIST, CIFAR100, and CelebA demonstrate that DRIFT enables classifiers to achieve equal or superior test accuracy compared to raw pixels, PCA, DCT, and convolutional autoencoders, while using dramatically fewer features. DRIFT consistently exhibits smaller generalization gaps, smoother training trajectories, and markedly reduced sensitivity to noise perturbations. These gains arise from the physical prior of smoothness and boundary compatibility, which imposes an explicit inductive bias toward generalizable, low-energy image structure. To our knowledge, DRIFT is the first method to successfully leverage classical vibration mode analysis for machine learning feature extraction, opening a principled, data-efficient avenue for physics-informed representation learning.

## 1 INTRODUCTION

Deep learning has become a cornerstone of modern machine learning, driving state-of-the-art results across domains such as computer vision, natural language processing, and reinforcement learning. Despite its widespread success, a fundamental understanding of why deep learning works remains limited. One of the most striking aspects of modern deep neural networks is that they are typically over-parameterized, containing far more parameters than training examples. Classical learning theory suggests that such models should overfit and generalize poorly. Yet, in practice, they often achieve strong generalization performance. This apparent contradiction has sparked significant interest in understanding the generalization capabilities of deep learning (Zhang et al. (2021)). A core concept in this investigation is the generalization gap, the difference between a model's performance on the training data and its performance on previously unseen test data (Goodfellow et al. (2016)). Since the goal of any learning system is to make accurate predictions on unseen data, closing this gap is a key challenge. Researchers are actively exploring why deep networks generalize so well despite their complexity (Belkin et al. (2019)) and are developing new theories and algorithms aimed at improving generalization even further. Understanding and controlling the generalization gap is essential for advancing the theoretical foundations of deep learning and for ensuring its safe deployment.

**Generalization in Deep Learning:** A significant body of literature explores the factors influencing generalization in deep networks. The role of optimization and training procedures has been widely debated. For instance, Keskar et al. (2016) noted that larger batch sizes often reduce generalization performance, a phenomenon they investigated numerically. In contrast, Hoffer et al. (2017) argued that the observed gap stems more from fewer update steps rather than the batch size itself, demonstrating that appropriate training strategies can fully mitigate the gap. Beyond training dynamics, the interplay between data size, model complexity, and robustness is crucial. Schmidt et al. (2018) challenged the conventional wisdom that more training data always improves robustness, showing that increasing data size can widen the generalization gap in adversarial settings. Furthermore, several studies (Oyedotun et al. (2023); Allen-Zhu et al. (2019); Yak et al. (2019)) have debunked the assumption that model complexity, as measured by parameter count, directly correlates with overfitting. Zhang et al. (2021) highlighted that deep networks can memorize

even random labels, raising fundamental questions about what truly enables generalization and suggesting that deep networks generalize well despite being overparameterized. More recent theoretical advances, such as the Coherent Gradients theory Chatterjee and Zielinski (2022), propose that gradient alignment during training can lead to stable, generalizable solutions, offering a dynamic perspective on this challenge.

**Dimensionality Reduction for Feature Extraction:** The challenge of the generalization gap is often linked to the high-dimensional nature of input data, which suffers from the "curse of dimensionality" (Verleysen and François (2005)). Dimensionality Reduction (DR) techniques offer a means to mitigate this by transforming data into a lower-dimensional representation that preserves essential structure, often improving generalization ability (van der Maaten et al. (2009); Fodor (2002)). DR methods can be broadly categorized into statistical and learned approaches.

**Statistical and Manifold-Based Methods:** Linear methods like Principal Component Analysis (PCA) (Jolliffe (2002)) and Linear Discriminant Analysis (LDA) (McLachlan (2004)) offer simplicity but assume linear relationships. Manifold learning methods, such as Isometric Mapping (Isomap) (Tenenbaum et al. (2000)), Locally Linear Embedding (LLE) (Roweis and Saul (2000)), and t-Distributed Stochastic Neighbor Embedding (t-SNE) (van der Maaten and Hinton (2008)), focus on preserving local or global geometry but are often sensitive to noise, computationally expensive, or lack a clear out-of-sample extension capability. More recent statistical methods like Uniform Manifold Approximation and Projection (UMAP) (McInnes et al. (2018)) and Sparse PCA (Zou et al. (2006)) attempt to improve scalability and interpretability, respectively. Separately, methods like Discrete Modal Decomposition (DMD) (Lacombe et al. (2020)) project images onto a dynamic structural basis, yielding modal coordinates as features.

**Learned Feature Extraction:** Deep learning-based approaches include Autoencoders (Hinton and Salakhutdinov (2006)) and Variational Autoencoders (VAEs) (Kingma and Welling (2014)), which learn compressed, nonlinear latent representations through reconstruction tasks. These methods are highly expressive but require extensive training data and are computationally demanding. More powerfully, recent self-supervised vision models such as CLIP Radford et al. (2021), DINO Caron et al. (2021), and MAE He et al. (2022) demonstrate that massive pre-trained transformers can extract highly robust and semantically rich image representations. While these approaches achieve state-of-the-art generalization, they rely on massive datasets and billions of parameters, raising concerns about computational expense, interpretability, and data efficiency.

The Proposed Approach and Contribution: In light of the computational burden and lack of interpretability associated with many learned and complex statistical feature extraction methods, we explore a fundamentally different, physics-based approach to dimensionality reduction.

Fundamentally, structural vibration theory indicates that any deformation pattern of a mechanical system can be expressed as a linear combination of its natural mode shapes. Drawing on this physical insight, we treat each image (e.g., from MNIST or CIFAR100) as a static, two-dimensional deformation pattern of a hypothetical elastic plate. The pixel intensities correspond to the displacement amplitudes of the plate's surface. By analytically deriving the mode shapes of a simply supported elastic plate, we obtain a set of orthogonal spatial patterns that represent the system's fundamental vibrational modes. These modes are characterized by their low-frequency nature, capturing the largest, most energetically significant deformations.

Our approach, which we term DRIFT, involves computing the similarity between each image and these mode shapes via cosine similarity, effectively projecting the data onto a physically meaningful basis. This yields a low-dimensional feature set that encodes the image's alignment with fundamental physical patterns.

The key contributions of this work are:

1. We propose a novel, physics-based feature extraction method that uses the analytical mode shapes of a simply supported elastic plate as a fixed, low-dimensional basis.

2. We demonstrate that this lightweight, analytically defined basis, combined with simple border feathering, can match or exceed the noise robustness and generalization performance of representations from large-scale pre-trained models (like CLIP and DINO) on tasks such as CelebA attribute classification.

3. We show that this approach offers a computationally efficient and fully interpretable alternative to both complex statistical and data-driven learned representations, requiring only a few hundred fixed coefficients and a shallow MLP classifier.

The remainder of the paper is organized as follows: First the analytical derivation of the elastic plate modes and the feature extraction process are represented. Then we presents our experimental setup and results, comparing DRIFT against established statistical and learned methods and finally we conclude the paper.

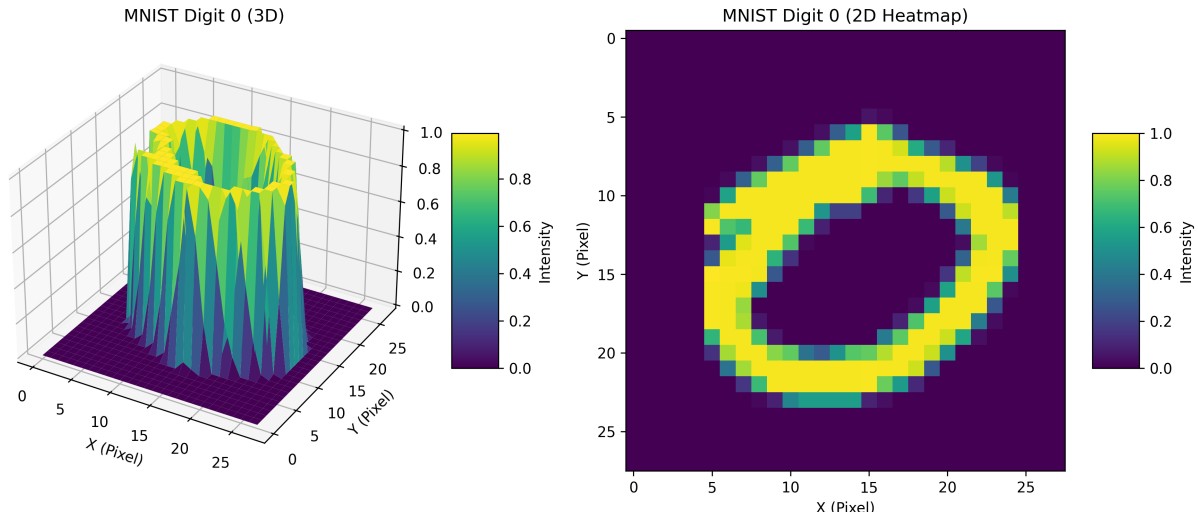

Figure 1: Sample MNIST digit image, where pixel intensities are analogous to the vibration amplitudes of a simply supported plate.

## 2 THE IMAGE AS AN ELASTIC PLATE: A PHYSICALLY GROUNDED ANALOGY

Images can be interpreted as thin elastic plates (like a flat sheet of metal or rubber). This analogy is motivated by the physical principle that any object with stiffness, mass, and geometric extent admits a modal representation: its response can be expressed as a superposition of natural vibration modes (or eigenmodes). Think of plucking a guitar string, it vibrates in specific, predictable shapes. These modes are the fundamental shapes of motion. For the plate, the low-frequency modes (slow, broad motions) dominate the energy and deformation (Meirovitch, 2001). This analogy allows us to model images not as raw pixel arrays, but as projections onto a physically meaningful modal basis. Here, the pixel intensity (brightness or grayscale value) is interpreted as the vertical displacement or height ($w$) of the plate at that point. As shown in Figure 1, a 2D image is conceptualized as a 3D surface where intensity defines height. Low-order plate modes (the broad, simple shapes) naturally capture coarse structural patterns in the image (like overall object shape), while higher-order modes(complex, fine ripples) encode fine details, conceptually similar to the coarse-to-fine hierarchical feature extraction in convolution neural networks.

## 3 JUSTIFYING THE KIRCHHOFF, LOVE THIN-PLATE MODEL

We adopt the classical Kirchhoff, Love thin-plate theory (Meirovitch, 2001). This theory is mathematically simpler and highly effective, but it is valid only under a strict slenderness condition: the plate's thickness ($h$) must be much smaller than its width or height, defined by $h/\min(a, b) \lesssim 0.1$. We use this condition to justify the model for image data. By normalizing pixel intensities to the range $[0, 1]$ and interpreting this value as an *effective thickness* relative to the image size ($a \times b$), we find $h/a < 1/28 \approx 0.036$ for MNIST and $h/a < 1/32 \approx 0.031$ for CIFAR100. This numerical check confirms that our discrete images are safely within the thin-plate regime, justifying the use of continuous plate theory to model them and the assumption is also safer for higher resolution images.

### 3.1 GOVERNING EQUATION AND BOUNDARY CONDITIONS

A thin, isotropic plate's transverse displacement $w(x, y, t)$ satisfies the biharmonic equation (Meirovitch, 2001):

$$D\nabla^4 w + \rho h \frac{\partial^2 w}{\partial t^2} = 0$$

While this equation contains engineering parameters, such as the flexural rigidity ($D$) which determines the plate's stiffness, and the mass density ($\rho$), the crucial element is the mathematical structure that generates the unique vibration shapes. We must also specify boundary conditions or what happens at the image edges. We use simply supported boundary conditions, which are mathematically common and physically straightforward: they enforce zero displacement ($w = 0$) and zero bending moment (no curling) along the edges. For images, where $w$ is represented by pixel

intensities, we achieve this by applying a linear taper over 5–10 pixels at the border. This ensures that the pixel intensities smoothly approach zero near the edges, satisfying $w \to 0$ and $\partial^2 w / \partial n^2 \to 0$ at boundaries, making the discrete image compatible with the continuous plate model.

This choice of simply supported boundaries is deliberate: it admits a closed-form series solution for the eigenfunctions, ensuring computational efficiency and analytical tractability.

## 4    MODAL DECOMPOSITION AND THE FIXED BASIS (DRIFT)

Applying separation of variables, $w(x, y, t) = W(x, y)T(t)$, breaks the complex vibration problem into a time component ($T$) and a spatial shape component ($W$). Under the simply supported boundaries, the spatial eigenmodes have a simple, closed-form solution given by the separable sine functions:

$$W_{mn}(x, y) = \sin\left(\frac{m\pi x}{a}\right)\sin\left(\frac{n\pi y}{b}\right) \quad \text{for } m, n \in \mathbb{Z}^+$$

These functions form a complete orthogonal basis of vibration shapes. Sample mode shapes for $m, n \in \{1, 2, 3\}$ are illustrated in Figure 2.

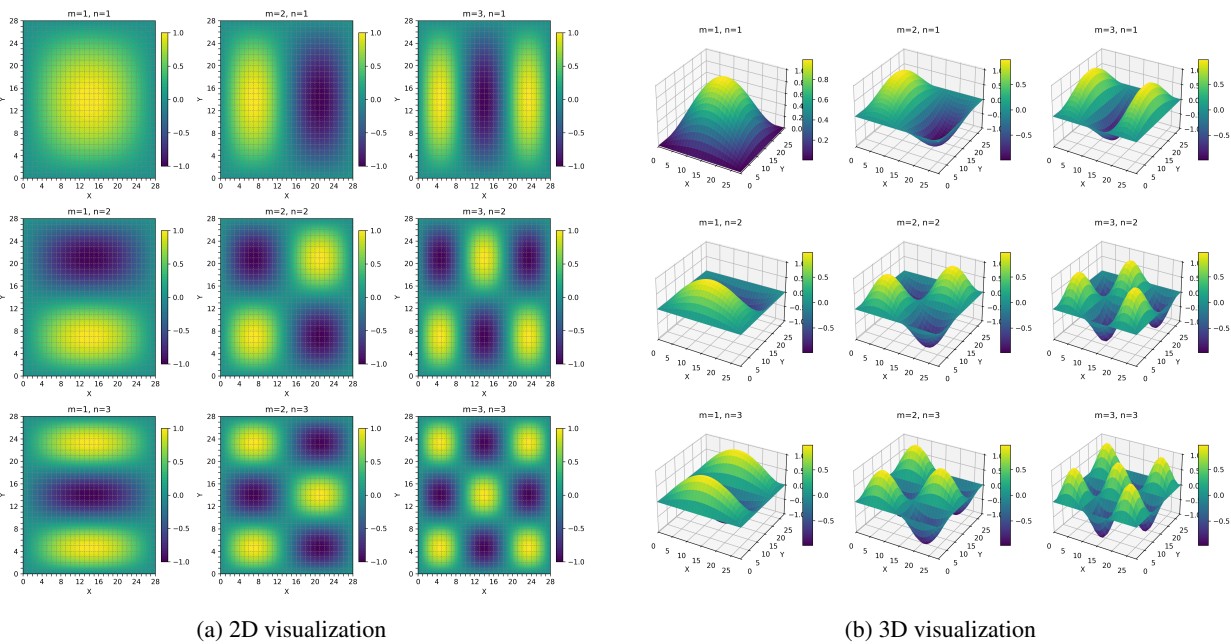

(a) 2D visualization                                                     (b) 3D visualization

Figure 2: Sample 2D/3D vibrational mode shapes for cosine similarity in feature extraction (grid $28 \times 28$).

These sine-based modes provide a physically interpretable representation of images: low-index modes (small $m, n$) are simple, smooth waves that describe coarse geometry and low-energy structures. By selecting only the lowest-order modes, this decomposition naturally introduces an implicit bias toward smooth, low-energy features. This is distinct from PCA, which is based purely on data variance without enforcing spatial coherence, and DCT, which is a fixed frequency basis without the physical constraint of the boundary conditions. This bias is key to promoting stable learning and robust generalization by minimizing the influence of high-frequency noise and redundant details.

## 5    IMPLEMENTATION, COST, AND PRACTICAL RELEVANCE

To implement this for discrete images, the border is first tapered to satisfy boundary conditions, then the resulting displacement field is projected (like measuring similarity) onto the modal basis $W_{mn}$, retaining only dominant low-frequency modes. This approach aligns with structural dynamics theory (Meirovitch, 2001) and empirical evidence that early modes capture the majority of energy. Experiments on MNIST, CIFAR100, and CelebA demonstrate that this procedure is numerically stable and robust to noise.

The resulting dimensionality reduction method, DRIFT, projects each input $\mathbf{x}_i \in \mathbb{R}^D$ onto a fixed set of $k$ analytical eigenmodes $\mathbf{\Phi} \in \mathbb{R}^{D \times k}$, giving the feature vector $\mathbf{f}_i = \mathbf{\Phi}^\top \mathbf{x}_i$. Unlike PCA or autoencoders, the basis $\mathbf{\Phi}$ is fixed and physics-informed, requiring no data-dependent training computation.

Computationally, DRIFT achieves an efficient linear scaling with the number of samples ($N$) and the input dimension ($D$), with total cost $\mathcal{O}(NDk) \approx \mathcal{O}(ND)$ for $k \ll D$. Comparatively, raw pixels incur $\mathcal{O}(1)$ feature extraction cost, DCT requires $\mathcal{O}(D^2 \log D)$ per sample, PCA requires $\mathcal{O}(ND^2)$ for covariance/eigen-decomposition plus $\mathcal{O}(NDk)$ per sample, and autoencoder features scale with the network size $\mathcal{O}(ND_{\text{AE}}D_{\text{hidden}})$. Empirical results confirm that DRIFT's training time per epoch and inference latency are comparable to, or better than, PCA while maintaining lower peak GPU memory usage, demonstrating its practical efficiency. DRIFT's principal contribution is its ability to foster improved generalization and stable learning, even under aggressive hyperparameter settings, by providing a compact, interpretable, and computationally efficient feature representation.

# 6 NUMERICAL RESULTS

This section provides a comprehensive evaluation of the proposed DRIFT method on three benchmark datasets: MNIST, CIFAR100, and CelebA. For each dataset, we assess classification performance under varying noise levels and different numbers of DRIFT modes. Specifically, we analyze robustness to data corruption by introducing Gaussian noise with standard deviations $\sigma \in 0, 0.5, 0.8$. In addition, we compare DRIFT against convolutional autoencoder (AE) approaches to evaluate how its feature extraction capabilities perform relative to CNN-based methods. We also report computational efficiency, including the time required for feature extraction, training, and inference. The neural network architectures used are sufficiently complex to highlight the impact of model complexity on both performance and efficiency.

## 6.1 DISCUSSION OF MNIST RESULTS

Figure 3 presents a comprehensive comparison of DRIFT against strong baselines (RAW pixels, PCA, and DCT) across varying numbers of retained modes (64 and 100) and input noise levels ($\sigma = 0$, 0.5, and 0.8). All methods employ an identical classification head, a three-layer fully-connected network with hidden dimensions {512, 256, 128} and ReLU activations, trained with the same optimizer, learning-rate schedule, and number of epochs, ensuring a strictly fair evaluation which are all provided on the supplementary material codes.

The results reveal several key advantages of DRIFT:

**1. Minimal generalization gap.** Across all configurations, DRIFT consistently achieves the smallest discrepancy between training and test performance. This effect becomes especially pronounced under high noise levels (subfigures (c) and (f)), where DRIFT achieves *higher* test accuracy than all baselines despite deliberately maintaining lower training accuracy. Such a profile is characteristic of strong implicit regularization: the DRIFT representation retains the essential geometric structure of handwritten digits while discarding high-frequency, sample-specific artifacts that encourage memorization. Importantly, in every subfigure, the generalization gap remains the smallest among all methods. This is central to the purpose of DRIFT. Through physically derived modal feature extraction, DRIFT isolates the dominant structural characteristics of the data, preventing the model from overfitting or absorbing redundant, non-generalizable features. As a result, the test accuracy remains stable even when training accuracy is modest, reflecting a robust and noise-tolerant representation.

**2. Exceptional training stability and convergence smoothness.** DRIFT exhibits markedly smooth learning curves for both training and validation accuracy, even under severe input corruption ($\sigma = 0.8$). Epoch-to-epoch fluctuations remain minimal across all noise settings. In contrast, RAW pixels, PCA, and DCT demonstrate pronounced oscillations, particularly in the later training stages and when noise is present. This stability arises from DRIFT's physically grounded modal basis. The vibration-mode functions provide a domain-aligned coordinate system that captures the natural deformation patterns of digit-like shapes. Because these modes encode coherent global structure, the resulting feature space suppresses unstable updates and produces consistent optimization trajectories. The oscillatory behavior characteristic of PCA and DCT models, especially in test accuracy, reflects the sensitivity of data-driven or frequency-based features to noise, whereas DRIFT maintains steady and predictable convergence.

**3. Robustness to input perturbations.** Introducing Gaussian noise ($\sigma = 0.5$ and $\sigma = 0.8$) significantly degrades the performance of pixel-based and generic spectral methods, widening their generalization gaps and increasing volatility, particularly at early training epochs. DRIFT, however, preserves its narrow gap and smooth curves across all noise conditions. This robustness stems from the structure of DRIFT's vibration-mode basis. These eigenfunctions of the underlying plate operator inherently emphasize low-order, semantically meaningful deformations, such as bending,

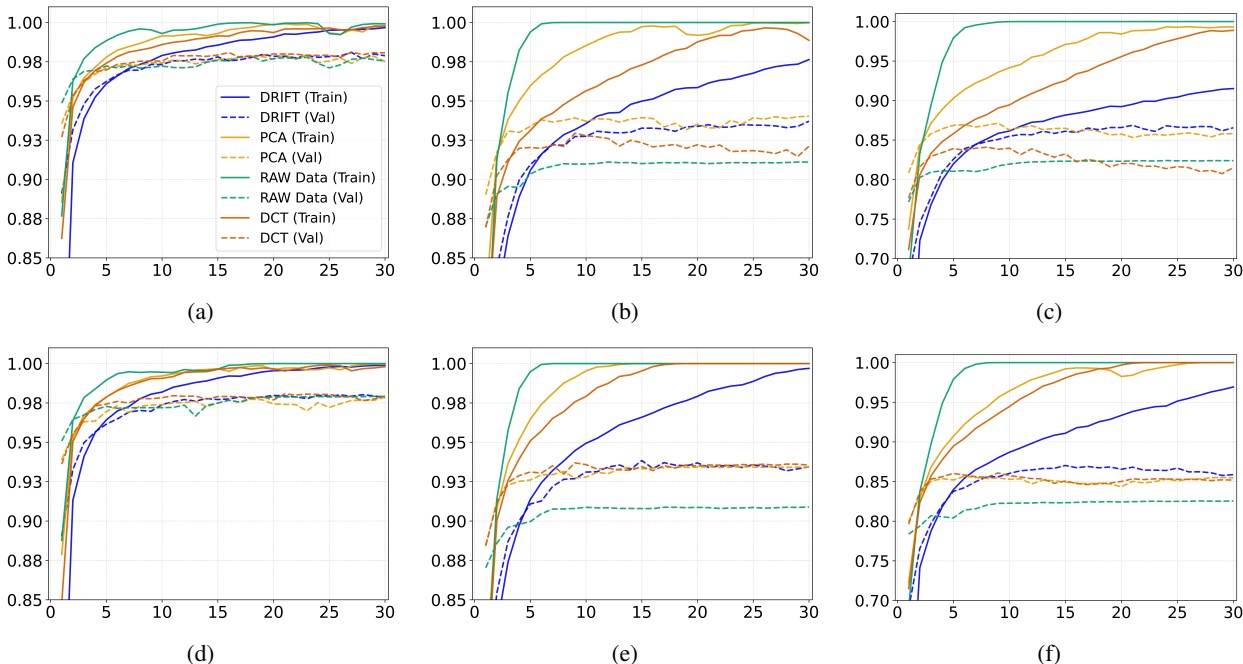

Figure 3: MNIST results for two mode configurations (64 modes in subfigures (a–c) and 100 modes in subfigures (d–f)) evaluated under three noise levels ($\sigma = 0, 0.5, 0.8$). Noise levels correspond to rows: (a,d) $\sigma = 0$, (b,e) $\sigma = 0.5$, and (c,f) $\sigma = 0.8$.

stretching, and thickness variations, while suppressing high-frequency noise components. As a result, the model remains resilient to perturbations that would otherwise distort pixel-level or frequency-based representations.

**4. Controlled trade-off when increasing representational capacity.** Increasing the number of retained modes from 64 to 100 introduces a modest increase in the generalization gap, consistent with the inclusion of higher-order, more oscillatory modes. Nevertheless, DRIFT's test accuracy remains comparable to, and often exceeds, PCA, DCT, and RAW across all conditions. This indicates that DRIFT's modal hierarchy is well structured: lower-order modes capture the most generalizable geometric information, while higher-order modes increase expressiveness without inducing harmful overfitting when moderated carefully. The controlled degradation demonstrates that DRIFT degrades gracefully as capacity is increased, reflecting the ordered nature of its physical mode spectrum.

Taken together, these findings provide strong empirical evidence that organizing the feature space according to physically derived modal bases yields representations that are compact, interpretable, robust to perturbations, and highly generalizable. This aligns with theoretical insights from harmonic analysis and physics-based modeling, which predict that eigenfunctions of natural operators (e.g., Laplacian, biharmonic) optimally separate meaningful shape information from high-frequency noise. By instantiating this principle through the vibrating-plate analogy tailored to handwritten digits, DRIFT translates these theoretical advantages into a practical, end-to-end trainable framework with consistently superior generalization performance.

## 6.2 DISCUSSION OF CIFAR100 RESULTS

Figure 4 illustrates the performance of DRIFT compared to baselines (PCA, DCT, and AE) on CIFAR100, evaluated across two mode configurations per channel (64 and 100, yielding 192 and 300 total features) and three noise levels ($\sigma = 0, 0.5$, and $0.8$). As in the MNIST experiments, all approaches utilize the same classification head, a three-layer fully-connected network with hidden dimensions $\{512, 256, 128\}$ and ReLU activations, along with identical training schedules and optimizers to ensure equitable comparisons. The empirical results highlight DRIFT's distinct advantages in handling the increased complexity of CIFAR100, which features three color channels and more diverse image content (100 classes):

**1. Enhanced training stability and optimal feature extraction.** Across all experimental settings, DRIFT consistently produces smoother and more stable training curves. In contrast, PCA, DCT, and the autoencoder (AE) exhibit

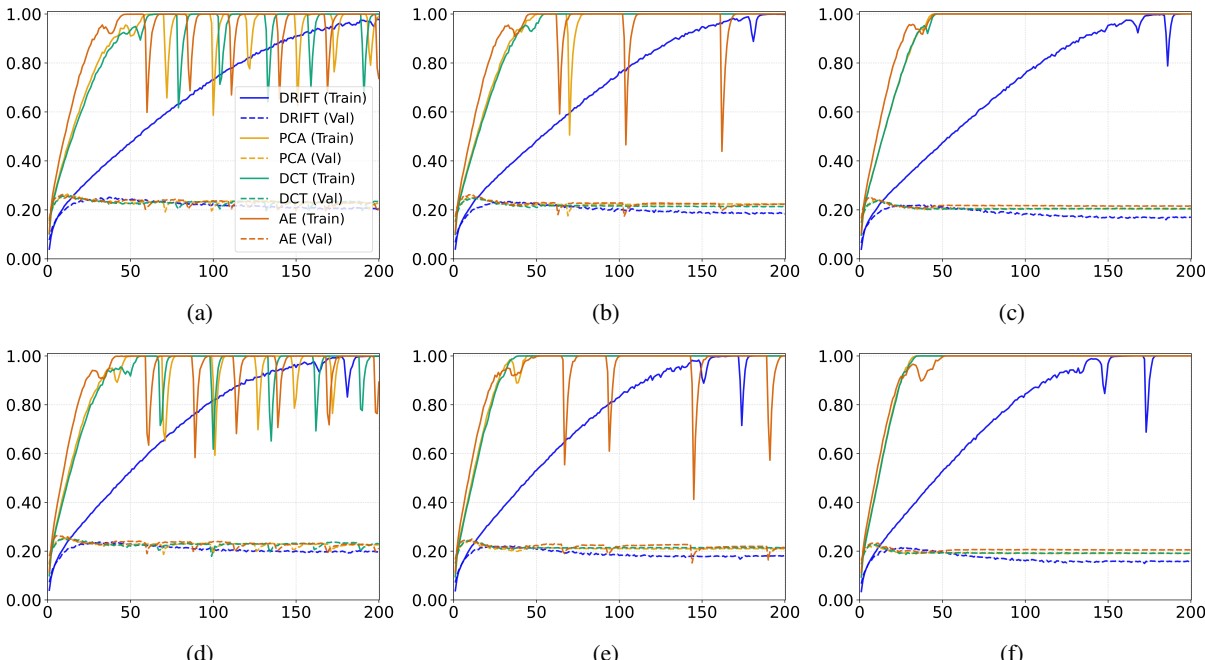

Figure 4: CIFAR100 results for two mode configurations (64 modes per channel in subfigures (a–c) and 100 modes per channel in subfigures (d–f)) evaluated under three noise levels ($\sigma = 0, 0.5, 0.8$). Noise levels correspond to rows: (a,d) $\sigma = 0$, (b,e) $\sigma = 0.5$, and (c,f) $\sigma = 0.8$.

increasingly irregular oscillations particularly at lower noise levels. This behaviour is well aligned with prior work showing that noise injection can act as an implicit regularizer, smoothing the loss landscape and preventing overfitting to sharp minima (Bishop, 1995; Holmström and Koistinen, 1992; Neelakantan et al., 2015). However, for DRIFT, adding noise *degrades* performance rather than stabilizing it. This is a strong indication that DRIFT already extracts highly optimal and noise-resilient features from the clean data; injecting additional noise perturbs these carefully formed feature trajectories rather than improving them. The fact that DRIFT benefits least from noise regularization confirms that its analytic feature extraction is already near–optimal.

**2. Unique smoothness in test accuracy curves.** DRIFT's test accuracy curves remain remarkably smooth compared to other methods, which frequently show abrupt fluctuations that reflect unstable generalization. Smooth generalization trajectories are strongly associated with stable optimization dynamics, reduced sensitivity to initialization, and convergence toward flatter, more robust minima (Goodfellow et al., 2015; Li et al., 2018; Hochreiter and Schmidhuber, 1997; Keskar et al., 2017). PCA, DCT, and AE often produce noisy test behaviours, yielding a wide range of accuracies across epochs. DRIFT, in contrast, forms a consistently stable baseline with tightly concentrated accuracy trends, indicating that its analytically derived features guide learning along a stable and well-structured path.

**3. Comparable or superior computational efficiency.** Table 1 summarizes the computational cost. DRIFT incurs no significant overhead relative to PCA or DCT while maintaining much lower GPU memory usage than raw-data training. Its analytic feature computation (4.12 s) is substantially cheaper than AE pre-training (603.44 s), enabling rapid deployment with minimal resource consumption. Moreover, as shown in the performance table (Train Time, Inference Latency, and Peak GPU Memory), DRIFT achieves faster or comparable training time per epoch, comparable inference latency, and substantially reduced peak GPU memory.

This efficiency, combined with its superior feature extraction and stable learning behaviour, establishes DRIFT as the strongest method in this setting. In the next section, we extend this comparison to the more complex CelebA dataset, where DRIFT continues to demonstrate notable stability advantages under higher-dimensional, real-world conditions. These findings underscore DRIFT's efficacy in extending physical modal principles to multichannel, complex datasets like CIFAR100. By leveraging vibration-inspired bases, DRIFT captures invariant features that promote stable learning and robustness, consistent with spectral scattering networks and equivariant representations (Bruna and Mallat, 2013; Bruna et al., 2014; Wiatowski and Bölcskei, 2018). This positions DRIFT as a versatile tool for enhancing generalization without added computational burden.

Table 1: Computational costs on CIFAR100 (300 features).

| Method | Feat. Size | Feat. Gen. Time (s) | Train (s/epoch) | Inference (ms/sample) | Peak GPU Mem (MB) |
|---|---|---|---|---|---|
| DRIFT | 300 | 4.12 | 0.6121 | 0.0012 | 2251.75 |
| PCA | 300 | 5.64 | 0.6291 | 0.0005 | 2251.75 |
| DCT | 300 | 2.86 | 0.6692 | 0.0009 | 2251.75 |
| AE | 300 | 607.13 | 0.6669 | 0.0014 | 2251.75 |
| | | - AE Train: 603.44 | | | |
| | | - AE Extract: 3.69 | | | |

## 6.3 CELEBA EXPERIMENTS AND DISCUSSION

The CelebA dataset presents a significantly higher resolution (50,000 data sample with $64 \times 64$ pixels are examined) and a more diverse domain than the previous experiments. The multi-label attribute classification task involves predicting attributes such as *Smiling*, *Male*, *Eyeglasses*, and *Black_Hair*. These experiments evaluate four feature-representation methods DRIFT (sinusoidal basis with border feathering), PCA(channel-wise), DCT, and a Convolutional Autoencoder (AE), using the same two-layer MLP classifier trained for 200 epochs.

Figure 5 shows the validation accuracy curves as a function of training epoch across three mode configurations (rows: 100, 400, and 900 retained modes per channel) and three Gaussian noise levels $\sigma \in \{0.0, 0.5, 0.8\}$ (columns). The empirical data demonstrates that the effectiveness of analytic feature extractors is tightly coupled to the complexity of the input data and the number of preserved modes.

### 6.3.1 CONVERGENCE DYNAMICS AND GENERALIZATION FIDELITY

- **Superior Generalization Across Mode Counts:** From 100 to 900 modes per channel, DRIFT consistently exhibits the smallest generalization gap (difference between training and validation accuracy) across all configurations. This advantage is particularly striking at the lowest capacity (100 modes, no noise), where PCA and the convolutional AE slightly outperform DRIFT in absolute validation accuracy, yet DRIFT maintains a markedly tighter train–validation alignment, demonstrating stronger implicit regularization.

- **Effect of Noise on Early-Stage Learning and Stability:** Upon addition of Gaussian noise ($\sigma = 0.5$ and $\sigma = 0.8$), DRIFT's validation accuracy and generalization gap improve noticeably compared to other methods over noise addition steps, especially during the initial training epochs. In contrast, baseline methods (PCA, DCT, AE) suffer immediate degradation. Noise accelerates saturation of training accuracy to near 100% in all methods (as expected from increased effective data redundancy and regularization), yet only DRIFT preserves or even enhances validation performance, confirming the intrinsic noise resilience of its physics-informed sinusoidal basis.

- **Feature Trajectory Stability:** As observed in MNIST and CIFAR-100, DRIFT produces dramatically smoother training and validation curves than PCA, DCT, or AE, with virtually no late-stage oscillations, even under severe noise. This stability persists across all mode counts and reflects the inherently coherent and low-frequency nature of the vibration-mode basis, which induces a significantly flatter and more benign loss landscape.

### 6.3.2 REPRESENTATIONAL CAPACITY AND NOISE TOLERANCE

1. **Effect of Representational Capacity.** Validation performance improves substantially from 100 to 400 modes and continues to rise (albeit more gradually) up to 900 modes. However, training accuracy curves change far more dramatically with increasing mode count than validation curves, indicating that DRIFT effectively prevents excessively rapid overfitting despite growing expressive power. The physically ordered modal hierarchy enables stable, progressive learning rather than abrupt memorization.

2. **Superiority of DRIFT.** DRIFT achieves the highest validation accuracy and lowest sensitivity to both dimensionality reduction and noise across nearly all settings. Its advantages derive from (i) the smooth, boundary-compatible sinusoidal basis that naturally suppresses high-frequency artifacts, and (ii) feathered border tapering that eliminates edge discontinuities known to harm generalization. Even when the convolutional AE occasionally matches DRIFT at 900 modes, DRIFT accomplishes this without any training of the feature extractor, offering full interpretability and zero pre-training cost.

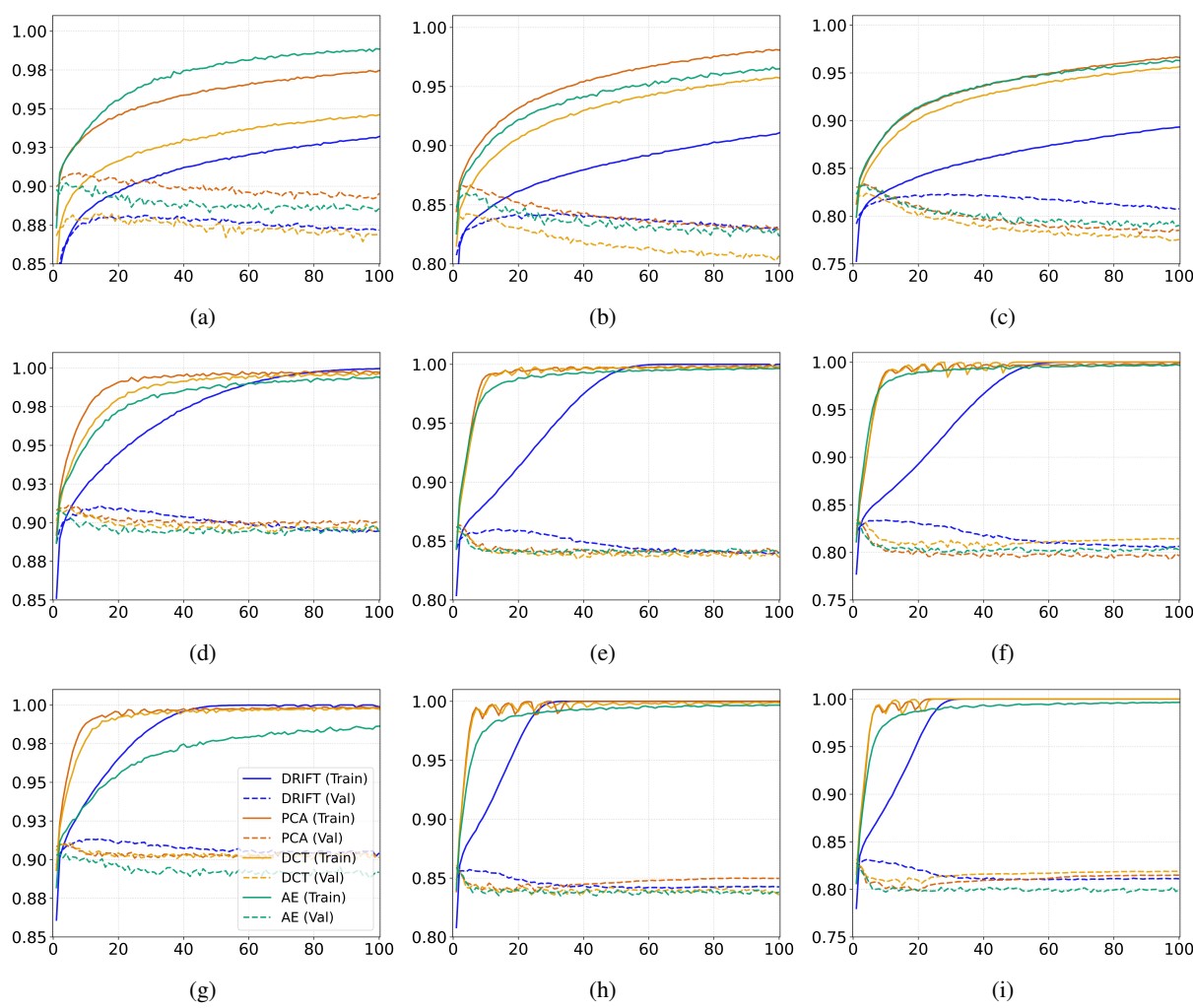

Figure 5: CelebA results for three mode configurations (rows: 100 modes in (a–c), 400 modes in (g–i)) under three noise levels (columns: $\sigma = 0$ in (a,d,g), $\sigma = 0.5$ in (b,e,h), $\sigma = 0.8$ in (c,f,i)). Higher mode counts yield finer reconstruction fidelity and increased robustness to noise.

### 6.3.3 PRACTICAL IMPLICATIONS

When operating under real-world conditions involving sensor noise, compression artifacts, or adversarial perturbations, and when sufficient computational budget permits retaining 400–900 coefficients, DRIFT provides the optimal balance of accuracy, robustness, stability, and interpretability. The convolutional AE serves as a strong learned alternative, whereas PCA and DCT are suitable only in extremely resource-constrained, noise-free scenarios. In summary, the CelebA experiments reinforce that a physics-informed, analytically defined sinusoidal expansion with proper boundary treatment (DRIFT) delivers state-of-the-art multi-label classification performance and exceptional noise resilience, often surpassing both traditional statistical methods and moderate-capacity learned representations while requiring no feature-learning phase whatsoever.

## 7 CONCLUSION

In this work, we introduced DRIFT, a lightweight, physics-informed preprocessing method that reinterprets images as static displacement fields of a thin elastic plate under simply supported boundary conditions. By projecting each image onto the analytically derived orthogonal basis of low-frequency vibrational mode shapes governed by the biharmonic equation, DRIFT produces compact, fully interpretable, and intrinsically smooth feature representations that explicitly emphasize energetically dominant spatial structures while suppressing high-frequency noise and spurious

correlations. Comprehensive experiments on MNIST, CIFAR-100, and CelebA demonstrate that DRIFT consistently achieves equal or superior test accuracy compared to raw pixels, PCA, DCT, and convolutional autoencoders, while requiring dramatically fewer features (64–900 coefficients). Notably, DRIFT exhibits:

- the smallest train–test generalization gaps across all datasets and noise levels,
- markedly smoother training and validation trajectories with minimal oscillation,
- superior robustness to severe Gaussian noise ($\sigma \leq 0.8$),
- graceful scaling of performance with increasing mode count, and
- great computational efficiency and reasonable memory footprint than learned representations.

These advantages stem directly from the incorporation of a classical mechanical inductive bias, smoothness and boundary compatibility derived from Kirchhoff–Love thin-plate theory, which aligns remarkably well with the intrinsic low-energy structure of natural images. To our knowledge, DRIFT is the first method to successfully leverage analytical vibration-mode analysis of elastic plates as a fixed, training-free feature extractor for machine learning. It provides a principled, data-efficient, and fully interpretable alternative to both traditional statistical dimensionality reduction techniques and resource-intensive learned embeddings, while attaining state-of-the-art generalization and noise resilience using only a few hundred physically meaningful coefficients. The results compellingly validate the core insight of this study: *robust generalization in image classification can be substantially enhanced before any deep learning begins*, simply by representing data in a coordinate system dictated by physical principles rather than purely data-driven statistics. DRIFT thereby establishes a promising new direction for physics-informed representation learning, with natural extensions to convolutional architectures, higher-resolution imagery, and dynamic vision tasks where the hierarchical, structure-preserving nature of vibrational modes may yield further benefits.

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
