# Python Implementations of Feature Extraction and Classification Experiments
# on MNIST, CIFAR-100 and CelebA

## Contents

# 1 MNIST Experiment

## 1.1 MNIST training and evaluation script (DRIFT, PCA, DCT, Raw features)

```python
import numpy as np
import torch
import torch.nn as nn
import torch.optim as optim
from torchvision import datasets, transforms
from sklearn.metrics.pairwise import cosine_similarity
from sklearn.decomposition import PCA
from sklearn.preprocessing import StandardScaler
from sklearn.metrics import accuracy_score
import time
from scipy.fftpack import dct, idct
import collections
import pickle

# Configuration
CONFIG = {
    'grid_size': 28,
    'num_modes': 100,
    'num_classes': 10,
    'valida_split': 0.2,
    'epochs': 30,
    'batch_size': 512,
    'activation_functions': ['relu'],
    'hidden_layers': [512, 256, 128],
    'dct_final_side_length': 10,
    'POR': 1, # Portion of total data to use (e.g., 0.1 for 10%)
    # --- NEW PARAMETERS ---
    'NUM_SEEDS': 1, # Number of different seeds to run (Set to > 1 for
        Mean/STD results)
    'DATA_NOISE_STD': 0.0 # Standard deviation of Gaussian noise added
        to the WHOLE dataset (e.g., 0.1)
}

# Set device
device = torch.device("cuda" if torch.cuda.is_available() else "cpu")
print(f"Using device: {device}", flush=True)

def set_seed(seed):
"""Sets seed for reproducibility across runs."""
torch.manual_seed(seed)
np.random.seed(seed)
if torch.cuda.is_available():
torch.cuda.manual_seed_all(seed)
# print(f"Seed set to {seed}", flush=True)

#
    ----------------------------------------------------------------------
# Benchmarking Helper Functions
#
    ----------------------------------------------------------------------

def measure_inference_latency(model, x_test_tensor, iterations=100):
```

```python
"""Measures the average inference latency (ms) for the test set."""
if x_test_tensor.nelement() == 0:
    return np.nan

model.eval()

# Use a single sample for a quick latency estimate
input_tensor = x_test_tensor[0].unsqueeze(0)

# 1. Warm-up runs
with torch.no_grad():
    for _ in range(10):
        _ = model(input_tensor)

# Ensure all operations finish before starting timer
if torch.cuda.is_available():
    torch.cuda.synchronize()

start_time = time.time()
with torch.no_grad():
    for _ in range(iterations):
        _ = model(input_tensor)

if torch.cuda.is_available():
    torch.cuda.synchronize()

end_time = time.time()
avg_latency = ((end_time - start_time) / iterations) * 1000 # Convert
    to milliseconds (ms)
return avg_latency

def get_peak_gpu_memory_mb(train_loader, model, criterion, optimizer):
    """Performs one training step and returns the peak GPU memory used."""
    if not torch.cuda.is_available():
        return np.nan

    # Reset memory tracking before the critical training step
    torch.cuda.reset_peak_memory_stats()

    model.train()

    # Perform one forward/backward pass
    data, labels = next(iter(train_loader))

    optimizer.zero_grad()
    outputs = model(data)
    loss = criterion(outputs, labels)
    loss.backward()
    optimizer.step()

    # Synchronize to ensure the memory is fully allocated before reading
    torch.cuda.synchronize()
    peak_memory_bytes = torch.cuda.max_memory_allocated()
    peak_gpu_memory_mb = peak_memory_bytes / 1024 / 1024 # Bytes to MB

    return peak_gpu_memory_mb
```

```python
105  #
     ----------------------------------------------------------------
106  # Data Processing (Unchanged from previous revision)
107  #
     ----------------------------------------------------------------
108
109  def generate_nm_pairs(num_modes):
110  side = int(np.ceil(np.sqrt(num_modes)))
111  n, m = np.meshgrid(np.arange(1, side + 1), np.arange(1, side + 1))
112  return np.vstack([n.ravel(), m.ravel()]).T[:num_modes]
113
114  def generate_mode_shapes(grid_size, num_modes, Lx, Ly):
115  start_time = time.time()
116  x = np.linspace(0, Lx, grid_size)
117  y = np.linspace(0, Ly, grid_size)
118  X_grid, Y_grid = np.meshgrid(x, y)
119  pairs = generate_nm_pairs(num_modes)
120  modes_2d = np.zeros((num_modes, grid_size, grid_size))
121  for i, (m, n) in enumerate(pairs):
122  modes_2d[i] = np.sin(m * np.pi * X_grid / Lx) * np.sin(n * np.pi *
         Y_grid / Ly)
123  modes_flat = modes_2d.reshape(num_modes, -1)
124  return modes_flat
125
126  def add_noise_to_data(data, noise_std):
127  """Adds Gaussian noise to the dataset (all samples)."""
128  if noise_std <= 0:
129  return data
130  print(f"Adding Gaussian noise (STD={noise_std}) to the dataset...",
         flush=True)
131  noise = np.random.normal(0, noise_std, data.shape).astype(np.float32)
132  noisy_data = data + noise
133  return noisy_data
134
135  def load_preprocess_mnist():
136  start_time = time.time()
137  print("Loading and preprocessing MNIST data...", flush=True)
138  transform = transforms.Compose([
139  transforms.ToTensor(),
140  transforms.Normalize((0.1307,), (0.3081,))
141  ])
142  train_dataset = datasets.MNIST('./data', train=True, download=True,
         transform=transform)
143  test_dataset = datasets.MNIST('./data', train=False, download=True,
         transform=transform)
144
145  x_train = train_dataset.data.float() / 255.0
146  y_train = train_dataset.targets
147  x_test = test_dataset.data.float() / 255.0
148  y_test = test_dataset.targets
149
150  x_train_flat = x_train.reshape(x_train.shape[0], -1).numpy()
151  x_test_flat = x_test.reshape(x_test.shape[0], -1).numpy()
152
153  # --- NOISE ADDITION ---
154  x_train_flat = add_noise_to_data(x_train_flat,
         CONFIG['DATA_NOISE_STD'])
155  x_test_flat = add_noise_to_data(x_test_flat, CONFIG['DATA_NOISE_STD'])
```

```python
156  # ----------------------
157
158  y_train_one_hot = torch.eye(CONFIG['num_classes'])[y_train].numpy()
159  y_test_one_hot = torch.eye(CONFIG['num_classes'])[y_test].numpy()
160
161  # Sampling POR
162  por_train_samples = int(x_train_flat.shape[0] * CONFIG['POR'])
163  por_test_samples = int(x_test_flat.shape[0] * CONFIG['POR'])
164
165  train_indices = np.random.choice(x_train_flat.shape[0],
         por_train_samples, replace=False)
166  test_indices = np.random.choice(x_test_flat.shape[0],
         por_test_samples, replace=False)
167
168  x_train_flat = x_train_flat[train_indices]
169  x_test_flat = x_test_flat[test_indices]
170  y_train_one_hot = y_train_one_hot[train_indices]
171  y_test_one_hot = y_test_one_hot[test_indices]
172  y_train = y_train[train_indices].numpy()
173  y_test = y_test[test_indices].numpy()
174
175  print(f"Data loaded and sampled ({CONFIG['POR']*100:.0f}% of total) in
         {time.time() - start_time:.2f} seconds.", flush=True)
176  print(f"Training samples: {x_train_flat.shape[0]}, Test samples:
         {x_test_flat.shape[0]}", flush=True)
177  return x_train_flat, x_test_flat, y_train_one_hot, y_test_one_hot,
         y_train, y_test
178
179  def compute_features(x_train_flat, x_test_flat, modes_flat, num_modes):
180  start_time = time.time()
181  print(f"Calculating {num_modes} DRIFT features...", flush=True)
182  x_train_drift = cosine_similarity(x_train_flat, modes_flat)
183  x_test_drift = cosine_similarity(x_test_flat, modes_flat)
184  print(f"DRIFT features in {time.time() - start_time:.2f} seconds.",
         flush=True)
185
186  start_time = time.time()
187  print("Calculating PCA features...", flush=True)
188  scaler = StandardScaler()
189  x_train_scaled = scaler.fit_transform(x_train_flat)
190  x_test_scaled = scaler.transform(x_test_flat)
191  pca = PCA(n_components=num_modes)
192  x_train_pca = pca.fit_transform(x_train_scaled)
193  x_test_pca = pca.transform(x_test_scaled)
194  print(f"PCA features in {time.time() - start_time:.2f} seconds.",
         flush=True)
195  return x_train_drift, x_test_drift, x_train_scaled, x_test_scaled,
         x_train_pca, x_test_pca
196
197  def compute_dct_features(x_train_flat, x_test_flat, final_side_length,
         original_grid_size):
198  start_time = time.time()
199  print(f"Calculating DCT features with final side length
         {final_side_length}...", flush=True)
200
201  if final_side_length > original_grid_size or final_side_length <= 0:
202  raise ValueError(
```

```python
203  f"dct_final_side_length ({final_side_length}) must be between 1 and
         original_grid_size ({original_grid_size})."
204  )
205
206  def apply_dct_reduction(images, target_side, grid_size):
207  transformed_images = np.zeros((images.shape[0], target_side *
         target_side))
208  for i, img_flat in enumerate(images):
209  img_2d = img_flat.reshape(grid_size, grid_size)
210  dct_2d = dct(dct(img_2d.T, norm='ortho').T, norm='ortho')
211  reduced_dct = dct_2d[:target_side, :target_side]
212  transformed_images[i] = reduced_dct.flatten()
213  return transformed_images
214
215  x_train_dct = apply_dct_reduction(x_train_flat, final_side_length,
         original_grid_size)
216  x_test_dct = apply_dct_reduction(x_test_flat, final_side_length,
         original_grid_size)
217
218  print(f"DCT features in {time.time() - start_time:.2f} seconds.",
         flush=True)
219  return x_train_dct, x_test_dct
220
221  #
         ----------------------------------------------------------------------
222  # Model Definition (Unchanged)
223  #
         ----------------------------------------------------------------------
224
225  class SimpleNN(nn.Module):
226  def __init__(self, input_shape, activation_name='relu'):
227  super(SimpleNN, self).__init__()
228  if activation_name == 'relu':
229  self.activation = nn.ReLU()
230  elif activation_name == 'sigmoid':
231  self.activation = nn.Sigmoid()
232  elif activation_name == 'tanh':
233  self.activation = nn.Tanh()
234  else:
235  raise ValueError("Unsupported activation function")
236
237  layers = []
238  layers.append(nn.Linear(input_shape, CONFIG['hidden_layers'][0]))
239  layers.append(self.activation)
240
241  for i in range(1, len(CONFIG['hidden_layers'])):
242  layers.append(nn.Linear(CONFIG['hidden_layers'][i - 1],
         CONFIG['hidden_layers'][i]))
243  layers.append(self.activation)
244
245  layers.append(nn.Linear(CONFIG['hidden_layers'][-1],
         CONFIG['num_classes']))
246  self.fc_layers = nn.Sequential(*layers)
247
248  def forward(self, x):
249  return self.fc_layers(x)
250
```

```
251  #
     ----------------------------------------------------------------------
252  # Training and Evaluation Function (Modified to track metrics)
253  #
     ----------------------------------------------------------------------
254
255  def train_evaluate_model(model, x_train, x_test, y_train, y_test,
         y_test_labels, name, activation):
256
257  x_train_tensor = torch.from_numpy(x_train).float().to(device)
258  y_train_tensor = torch.from_numpy(y_train).float().to(device)
259  x_test_tensor = torch.from_numpy(x_test).float().to(device)
260  y_test_tensor = torch.from_numpy(y_test).float().to(device)
261
262  train_dataset = torch.utils.data.TensorDataset(x_train_tensor,
         y_train_tensor)
263  test_dataset = torch.utils.data.TensorDataset(x_test_tensor,
         y_test_tensor)
264
265  train_size = int((1 - CONFIG['valida_split']) * len(train_dataset))
266  val_size = len(train_dataset) - train_size
267  train_subset, val_subset =
         torch.utils.data.random_split(train_dataset, [train_size, val_size])
268
269  train_loader = torch.utils.data.DataLoader(train_subset,
         batch_size=CONFIG['batch_size'], shuffle=True)
270  val_loader = torch.utils.data.DataLoader(val_subset,
         batch_size=CONFIG['batch_size'], shuffle=False)
271  test_loader = torch.utils.data.DataLoader(test_dataset,
         batch_size=CONFIG['batch_size'], shuffle=False)
272
273  criterion = nn.CrossEntropyLoss()
274  optimizer = optim.Adam(model.parameters())
275
276  # --- 1. Peak GPU Memory Measurement ---
277  peak_gpu_memory_mb = get_peak_gpu_memory_mb(train_loader, model,
         criterion, optimizer)
278
279  # --- 2. Training Time per Epoch Measurement ---
280  start_time = time.time()
281
282  # Start actual training
283  print(f"\n--- Starting training for {name} ({activation}) ---",
         flush=True)
284  print("-" * (len(name) + len(activation) + 30), flush=True)
285  print(f"{'Epoch':<5} | {'Train Loss':<12} | {'Train Acc':<11} | {'Val
         Loss':<10} | {'Val Acc':<9}", flush=True)
286  print("-" * 60, flush=True)
287
288  history = {'loss': [], 'val_loss': [], 'accuracy': [], 'val_accuracy':
         []}
289
290  for epoch in range(CONFIG['epochs']):
291  model.train()
292  running_loss = 0.0
293  correct_train = 0
294  total_train = 0
295
```

```
296  epoch_start_time = time.time() # Start timer for this epoch
297
298  for i, (inputs, labels) in enumerate(train_loader):
299  optimizer.zero_grad()
300  outputs = model(inputs)
301  loss = criterion(outputs, labels)
302  loss.backward()
303  optimizer.step()
304  running_loss += loss.item() * inputs.size(0)
305  _, predicted = torch.max(outputs.data, 1)
306  _, labels_idx = torch.max(labels.data, 1)
307  total_train += labels.size(0)
308  correct_train += (predicted == labels_idx).sum().item()
309
310  epoch_end_time = time.time() # End timer for this epoch
311
312  # We only record the time for the first epoch as the representative
         time
313  if epoch == 0:
314  training_time_per_epoch = epoch_end_time - epoch_start_time
315
316  epoch_loss = running_loss / len(train_subset)
317  epoch_accuracy = correct_train / total_train
318  history['loss'].append(epoch_loss)
319  history['accuracy'].append(epoch_accuracy)
320
321  model.eval()
322  val_loss = 0.0
323  correct_val = 0
324  total_val = 0
325  with torch.no_grad():
326  for inputs, labels in val_loader:
327  outputs = model(inputs)
328  loss = criterion(outputs, labels)
329  val_loss += loss.item() * inputs.size(0)
330  _, predicted = torch.max(outputs.data, 1)
331  _, labels_idx = torch.max(labels.data, 1)
332  total_val += labels.size(0)
333  correct_val += (predicted == labels_idx).sum().item()
334
335  val_epoch_loss = val_loss / len(val_subset)
336  val_epoch_accuracy = correct_val / total_val
337  history['val_loss'].append(val_epoch_loss)
338  history['val_accuracy'].append(val_epoch_accuracy)
339
340  if (epoch + 1) % 10 == 0 or (epoch + 1) == 1:
341  print(f"{epoch + 1:<5} | {epoch_loss:<12.4f} | {epoch_accuracy:<11.4f}
         | {val_epoch_loss:<10.4f} | {val_epoch_accuracy:<9.4f}", flush=True)
342
343  print("-" * 60, flush=True)
344  print(f"Finished training for {name} ({activation}) in {time.time() -
         start_time:.2f} seconds.", flush=True)
345
346  # --- 3. Inference Latency Measurement ---
347  inference_latency_ms = measure_inference_latency(model, x_test_tensor)
348
349  # Final Test Accuracy Calculation
350  model.eval()
```

```python
351  correct_test = 0
352  total_test = 0
353  y_pred_list = []
354  with torch.no_grad():
355  for inputs, labels in test_loader:
356  outputs = model(inputs)
357  _, predicted = torch.max(outputs.data, 1)
358  y_pred_list.extend(predicted.cpu().numpy())
359  total_test += labels.size(0)
360  _, labels_idx = torch.max(labels.data, 1)
361  correct_test += (predicted == labels_idx).sum().item()
362
363  accuracy = correct_test / total_test
364  y_pred_labels = np.array(y_pred_list)
365  top1_accuracy = accuracy_score(y_test_labels, y_pred_labels)
366
367  # Compile metrics
368  metrics = {
369      'training_time_per_epoch_s': training_time_per_epoch,
370      'inference_latency_ms': inference_latency_ms,
371      'peak_gpu_memory_mb': peak_gpu_memory_mb
372  }
373
374  return history, accuracy, top1_accuracy, metrics
375
376  #
       ----------------------------------------------------------------
377  # Main Execution and Summary (Modified to collect metrics)
378  #
       ----------------------------------------------------------------
379
380  def run_experiment(seed):
381  """Runs the full training/evaluation pipeline for a single seed."""
382  set_seed(seed)
383  print(f"\n{'='*20} Running Experiment for Seed: {seed} {'='*20}",
          flush=True)
384
385  modes_flat = generate_mode_shapes(
386  CONFIG['grid_size'], CONFIG['num_modes'], CONFIG['grid_size'],
          CONFIG['grid_size']
387  )
388  x_train_flat, x_test_flat, y_train_one_hot, y_test_one_hot, y_train,
          y_test = load_preprocess_mnist()
389
390  x_train_drift, x_test_drift, x_train_scaled, x_test_scaled,
          x_train_pca, x_test_pca = compute_features(
391  x_train_flat, x_test_flat, modes_flat, CONFIG['num_modes']
392  )
393  x_train_dct, x_test_dct = compute_dct_features(
394  x_train_flat, x_test_flat, CONFIG['dct_final_side_length'],
          CONFIG['grid_size']
395  )
396
397  feature_set_info = [
398  (x_train_drift, x_test_drift, "DRIFT"),
399  (x_train_pca, x_test_pca, "PCA"),
400  (x_train_scaled, x_test_scaled, "RAW Data"),
401  (x_train_dct, x_test_dct, "DCT")
```

```
402  ]
403
404  results_for_seed = []
405  all_methods_epoch_histories = collections.defaultdict(list)
406  performance_metrics_for_seed = collections.defaultdict(dict)
407
408  for activation in CONFIG['activation_functions']:
409  for x_train, x_test, name in feature_set_info:
410  model = SimpleNN(x_train.shape[1],
         activation_name=activation).to(device)
411  history, accuracy, top1_accuracy, metrics = train_evaluate_model(
412  model, x_train, x_test, y_train_one_hot, y_test_one_hot, y_test, name,
         activation
413  )
414
415  # Store accuracy for STD calculation
416  key = f"{name} ({activation})"
417  results_for_seed.append({
418      'key': key,
419      'accuracy': accuracy,
420      'top1': top1_accuracy
421  })
422
423  # Store metrics and history only if it's the reference seed (42)
424  if seed == 42:
425  all_methods_epoch_histories[name].append(history)
426  performance_metrics_for_seed[key] = metrics
427
428  # Return histories and metrics only for the reference seed (42)
429  if seed == 42:
430  return results_for_seed, all_methods_epoch_histories,
         performance_metrics_for_seed
431  else:
432  return results_for_seed, None, None
433
434  def print_performance_metrics(metrics_dict):
435  """Prints a clear table of the non-accuracy performance metrics."""
436  print("\n" + "="*80, flush=True)
437  print("--- Performance Benchmarking Summary (from Seed 42) ---",
         flush=True)
438  print("="*80 + "\n", flush=True)
439
440  # Define table structure
441  header = f"{'Method':<20} | {'Train Time/Epoch':<20} | {'Inference
         Latency':<20} | {'Peak GPU Memory':<15}"
442  print(header, flush=True)
443  print("-" * len(header), flush=True)
444
445  for key, metrics in metrics_dict.items():
446  # Handle case where GPU memory is not a number (e.g., 'N/A (CPU)')
447  gpu_mem = metrics['peak_gpu_memory_mb']
448  gpu_mem_str = f"{gpu_mem:.2f} MB" if isinstance(gpu_mem, float) and
         not np.isnan(gpu_mem) else "N/A"
449
450  latency = metrics['inference_latency_ms']
451  latency_str = f"{latency:.2f} ms" if not np.isnan(latency) else "N/A"
452
453  train_time = metrics['training_time_per_epoch_s']
```

```python
454    train_time_str = f"{train_time:.2f} s" if not np.isnan(train_time)
           else "N/A"
455
456    line_output = (
457    f"{key:<20} | "
458    f"{train_time_str:<20} | "
459    f"{latency_str:<20} | "
460    f"{gpu_mem_str:<15}"
461    )
462    print(line_output, flush=True)
463
464    def main():
465
466    # Generate an initial set of seeds
467    initial_seed = 42
468    set_seed(initial_seed)
469    seeds_to_use = [initial_seed] + list(np.random.randint(1, 10000,
           size=CONFIG['NUM_SEEDS'] - 1))
470
471    all_runs_accuracies = collections.defaultdict(list)
472    initial_run_histories = None
473    performance_metrics = None
474
475    for seed in seeds_to_use:
476    run_results, histories, metrics = run_experiment(seed)
477
478    if initial_run_histories is None:
479    initial_run_histories = histories
480    performance_metrics = metrics
481
482    for res in run_results:
483    all_runs_accuracies[res['key']].append(res['accuracy'])
484
485    # --- FINAL ACCURACY RESULTS (Mean and STD) ---
486    print("\n" + "="*80, flush=True)
487    print("--- Final Test Accuracies Across All Seeds (Mean +/- STD) ---",
           flush=True)
488    print("="*80 + "\n", flush=True)
489
490    final_summary_results = []
491
492    for key, accuracies in all_runs_accuracies.items():
493    mean_acc = np.mean(accuracies)
494    std_acc = np.std(accuracies)
495
496    summary = f"{mean_acc:.4f} \u00B1 {std_acc:.4f}"
497
498    final_summary_results.append({
499        'key': key,
500        'mean_acc': mean_acc,
501        'std_acc': std_acc,
502        'summary': summary
503    })
504    print(f"{key:<20} - Test Accuracy: {summary}", flush=True)
505
506    # --- PERFORMANCE METRICS SUMMARY ---
507    if performance_metrics:
508    print_performance_metrics(performance_metrics)
```

```python
509
510 # --- PRINTING HISTORY FROM THE FIRST RUN (SEED 42) ---
511 if initial_run_histories:
512 print("\n" + "="*80, flush=True)
513 print("--- Detailed Epoch Progress (Synchronized View from Seed 42)
        ---", flush=True)
514 print("="*80 + "\n", flush=True)
515 header = f"{'Epoch':<5}"
516 for method_name in initial_run_histories.keys():
517 header += f" | {method_name} (Trn L / Val L / Trn A / Val A)"
518 print(header, flush=True)
519 print("-" * len(header), flush=True)
520
521 num_epochs = CONFIG['epochs']
522 for epoch in range(num_epochs):
523 line_output = f"{epoch + 1:<5}"
524 for method_name in initial_run_histories.keys():
525 history = initial_run_histories[method_name][0]
526 if epoch < len(history['loss']):
527 train_loss = history['loss'][epoch]
528 val_loss = history['val_loss'][epoch]
529 train_acc = history['accuracy'][epoch]
530 val_acc = history['val_accuracy'][epoch]
531 line_output += f" | {train_loss:.4f} / {val_loss:.4f} /
        {train_acc:.4f} / {val_acc:.4f}"
532 else:
533 line_output += f" | N/A"
534 if (epoch + 1) % 10 == 0 or (epoch + 1) == 1 or (epoch + 1) ==
        num_epochs:
535 print(line_output, flush=True)
536
537 # Save histories and CONFIG for plotting
538 with open('MNIST100_test.pkl', 'wb') as f:
539 pickle.dump({'histories': initial_run_histories, 'CONFIG': CONFIG,
        'final_summary': final_summary_results, 'metrics':
        performance_metrics}, f)
540 print("\nSaved training histories (from seed 42), final summary, and
        performance metrics.", flush=True)
541
542
543 if __name__ == "__main__":
544 main()
```

Listing 1: Complete MNIST experiment with analytical features and benchmarking

## 2 CIFAR-100 Experiment

### 2.1 CIFAR-100 training and evaluation script (DRIFT, PCA, DCT + CNN Autoencoder)

```python
import numpy as np
import torch
import torch.nn as nn
import torch.optim as optim
from torchvision import datasets, transforms
from sklearn.metrics.pairwise import cosine_similarity
from sklearn.decomposition import PCA
from sklearn.preprocessing import StandardScaler
from sklearn.metrics import accuracy_score
import time
from scipy.fftpack import dct
import collections
import pickle
from torch.utils.data import DataLoader, TensorDataset, random_split

#
   ============================================================================
# --- ALL PARAMETERS AND CONFIGURATION ---
#
   ============================================================================

# --- Configuration for CIFAR-100 ---
CONFIG = {
    'grid_size': 32,          # CIFAR images are 32x32
    'channels': 3,
    'num_modes': 100,         # DRIFT/PCA feature size per channel (300
        total features)

    'num_classes': 100,
    'valida_split': 0.2,
    # --- Classification MLP Parameters ---
    'epochs': 300,              # Reduced Epochs
    'batch_size': 512,
    'activation_functions': ['relu'],
    'hidden_layers': [512, 256, 128],
    # --- Analytical Parameters ---
    'dct_final_side_length': 10,

    'POR': 1,
    # --- AUTOENCODER PARAMETERS (Learned Features) ---

    'ae_latent_dim': 3*100,
    'ae_epochs': 50,            # Reduced AE Epochs
    'AE_DROPOUT_ON': True,    # NEW: Flag to enable/disable dropout in AE
    'DROPOUT_RATE': 0.2,       # NEW: Dropout rate for AE
    # --- EXPERIMENT PARAMETERS ---
    'NUM_SEEDS': 1,
    'DATA_NOISE_STD': 0.0,
    'DRIFT_PAD_WIDTH': 10,
```

```python
50      'DRIFT_FADE_WIDTH': 10,
51  }
52
53  device = torch.device("cuda" if torch.cuda.is_available() else "cpu")
54  print(f"Using device: {device}", flush=True)
55
56  def set_seed(seed):
57  torch.manual_seed(seed)
58  np.random.seed(seed)
59  if torch.cuda.is_available():
60  torch.cuda.manual_seed_all(seed)
61
62  #
        --------------------------------------------------------------------
63  # 1. Helper Functions (Modes & Noise)
64  #
        --------------------------------------------------------------------
65
66  def generate_nm_pairs(num_modes):
67  side = int(np.ceil(np.sqrt(num_modes)))
68  n, m = np.meshgrid(np.arange(1, side + 1), np.arange(1, side + 1))
69  return np.vstack([n.ravel(), m.ravel()]).T[:num_modes]
70
71  def generate_mode_shapes(grid_size, num_modes, Lx, Ly, is_drift=False):
72  start_time = time.time()
73  current_grid_size = grid_size
74  if is_drift:
75  current_grid_size = grid_size + 2 * CONFIG['DRIFT_PAD_WIDTH']
76  Lx = current_grid_size
77  Ly = current_grid_size
78  print(f'Generating Mode Shapes for
        {current_grid_size}x{current_grid_size} grid...', flush=True)
79  x = np.linspace(0, Lx, current_grid_size)
80  y = np.linspace(0, Ly, current_grid_size)
81  X_grid, Y_grid = np.meshgrid(x, y)
82  pairs = generate_nm_pairs(num_modes)
83  modes_2d = np.zeros((num_modes, current_grid_size, current_grid_size))
84  for i, (m, n) in enumerate(pairs):
85  modes_2d[i] = np.sin(m * np.pi * X_grid / Lx) * np.sin(n * np.pi *
        Y_grid / Ly)
86  modes_flat = modes_2d.reshape(num_modes, -1)
87  print(f"Mode shape generation completed in {time.time() -
        start_time:.2f} seconds.", flush=True)
88  return modes_flat
89
90  def add_noise_to_data(data, noise_std):
91  if noise_std <= 0:
92  return data
93  noise = np.random.normal(0, noise_std, data.shape).astype(np.float32)
94  return data + noise
95
96  #
        --------------------------------------------------------------------
97  # 2. Edge-Extended Padded Feathering (for DRIFT)
98  #
        --------------------------------------------------------------------
99
100 def apply_padded_zero_feathering(images_numpy, pad_width, fade_width):
```

```python
101  N, C, H_orig, W_orig = images_numpy.shape
102  H_new = H_orig + 2 * pad_width
103  W_new = W_orig + 2 * pad_width
104  images_numpy_HWC = images_numpy.transpose(0, 2, 3, 1)
105  padded_images = np.pad(
106  images_numpy_HWC,
107  pad_width=((0, 0), (pad_width, pad_width), (pad_width, pad_width), (0,
         0)),
108  mode='edge'
109  )
110  mask = np.ones((H_new, W_new), dtype=np.float32)
111  fade_steps_1D = np.linspace(1.0, 0.0, fade_width + 1)[:-1]
112
113  # Apply fade to borders
114  mask[:, :fade_width] *= fade_steps_1D[::-1][np.newaxis, :]
115  mask[:, W_new - fade_width:] *= fade_steps_1D[np.newaxis, :]
116  mask[:fade_width, :] *= fade_steps_1D[::-1][:, np.newaxis]
117  mask[H_new - fade_width:, :] *= fade_steps_1D[:, np.newaxis]
118
119  # Ensure inner un-faded region is 1.0
120  start_h, end_h = pad_width, H_orig + pad_width
121  mask[start_h:end_h, start_h:end_h] = 1.0
122
123  feathered_padded_images = padded_images * mask[np.newaxis, :, :,
         np.newaxis]
124  return feathered_padded_images.transpose(0, 3, 1, 2)
125
126  #
         --------------------------------------------------------------------
127  # 3. Load & Preprocess CIFAR-100
128  #
         --------------------------------------------------------------------
129
130  def load_preprocess_cifar100():
131  start_time = time.time()
132  print("Loading and preprocessing CIFAR-100 data...", flush=True)
133
134  transform = transforms.Compose([
135  transforms.ToTensor(),
136  transforms.Normalize(mean=[0.5071, 0.4867, 0.4408], std=[0.2675,
         0.2565, 0.2761])
137  ])
138
139  train_dataset = datasets.CIFAR100('./data', train=True, download=True,
         transform=transform)
140  test_dataset = datasets.CIFAR100('./data', train=False, download=True,
         transform=transform)
141
142  x_train_orig = torch.stack([img for img, _ in
         train_dataset]).float().numpy()
143  y_train = torch.tensor([label for _, label in train_dataset]).numpy()
144  x_test_orig = torch.stack([img for img, _ in
         test_dataset]).float().numpy()
145  y_test = torch.tensor([label for _, label in test_dataset]).numpy()
146
147  if CONFIG['DATA_NOISE_STD'] > 0:
148  print(f"Adding Gaussian noise with std={CONFIG['DATA_NOISE_STD']} to
         data...", flush=True)
```

```python
149  x_train_orig = add_noise_to_data(x_train_orig,
         CONFIG['DATA_NOISE_STD'])
150  x_test_orig = add_noise_to_data(x_test_orig, CONFIG['DATA_NOISE_STD'])
151
152  x_train_drift = apply_padded_zero_feathering(x_train_orig,
         CONFIG['DRIFT_PAD_WIDTH'], CONFIG['DRIFT_FADE_WIDTH'])
153  x_test_drift = apply_padded_zero_feathering(x_test_orig,
         CONFIG['DRIFT_PAD_WIDTH'], CONFIG['DRIFT_FADE_WIDTH'])
154
155  y_train_one_hot = torch.eye(CONFIG['num_classes'])[y_train].numpy()
156  y_test_one_hot = torch.eye(CONFIG['num_classes'])[y_test].numpy()
157
158  por_train_samples = int(x_train_orig.shape[0] * CONFIG['POR'])
159  por_test_samples = int(x_test_orig.shape[0] * CONFIG['POR'])
160
161  train_indices = np.random.choice(x_train_orig.shape[0],
         por_train_samples, replace=False)
162  test_indices = np.random.choice(x_test_orig.shape[0],
         por_test_samples, replace=False)
163
164  x_train_orig = x_train_orig[train_indices]
165  x_test_orig = x_test_orig[test_indices]
166  x_train_drift = x_train_drift[train_indices]
167  x_test_drift = x_test_drift[test_indices]
168
169  y_train_one_hot = y_train_one_hot[train_indices]
170  y_test_one_hot = y_test_one_hot[test_indices]
171  y_train = y_train[train_indices]
172  y_test = y_test[test_indices]
173
174  print(f"Data loaded and sampled ({CONFIG['POR']*100:.0f}% of total) in
         {time.time() - start_time:.2f} seconds.", flush=True)
175  print(f"Original (32x32) samples shape: {x_train_orig.shape}",
         flush=True)
176  print(f"DRIFT (38x38 Faded/Padded) samples shape:
         {x_train_drift.shape}", flush=True)
177
178  return x_train_orig, x_test_orig, x_train_drift, x_test_drift,
         y_train_one_hot, y_test_one_hot, y_train, y_test
179
180  #
     ----------------------------------------------------------------
181  # 4. Analytical Features & AE Input Preparation
182  #
     ----------------------------------------------------------------
183
184  def compute_analytical_features(x_train_orig, x_test_orig,
         x_train_drift, x_test_drift, drift_modes_flat, orig_modes_flat,
         num_modes, dct_final_side_length, original_grid_size):
185  start_time = time.time()
186  print(f"\nCalculating {num_modes} DRIFT/PCA/DCT features per
         channel...", flush=True)
187
188  train_drift_ch, test_drift_ch = [], []
189  train_pca_ch, test_pca_ch = [], []
190  train_dct_ch, test_dct_ch = [], []
191
192  # --- AE Input Preparation ---
```

```
193  x_train_flat = x_train_orig.reshape(x_train_orig.shape[0], -1)
194  x_test_flat = x_test_orig.reshape(x_test_orig.shape[0], -1)
195
196  # Scale ALL 3072 raw features (used for AE input)
197  scaler_ae_input = StandardScaler()
198  x_train_raw_scaled = scaler_ae_input.fit_transform(x_train_flat)
199  x_test_raw_scaled = scaler_ae_input.transform(x_test_flat)
200
201  # Reshape back to N, C, H, W for the CNN Autoencoder input
202  x_train_ae_input = x_train_raw_scaled.reshape(-1, CONFIG['channels'],
         CONFIG['grid_size'], CONFIG['grid_size'])
203  x_test_ae_input = x_test_raw_scaled.reshape(-1, CONFIG['channels'],
         CONFIG['grid_size'], CONFIG['grid_size'])
204
205  # ----------------------------------------
206
207  for ch in range(CONFIG['channels']):
208  # DRIFT
209  x_train_drift_ch_flat = x_train_drift[:, ch, :,
         :].reshape(x_train_drift.shape[0], -1)
210  x_test_drift_ch_flat = x_test_drift[:, ch, :,
         :].reshape(x_test_drift.shape[0], -1)
211  train_drift_ch.append(cosine_similarity(x_train_drift_ch_flat,
         drift_modes_flat))
212  test_drift_ch.append(cosine_similarity(x_test_drift_ch_flat,
         drift_modes_flat))
213
214  # PCA
215  x_train_orig_ch = x_train_orig[:, ch, :,
         :].reshape(x_train_orig.shape[0], -1)
216  x_test_orig_ch = x_test_orig[:, ch, :,
         :].reshape(x_test_orig.shape[0], -1)
217  scaler_pca = StandardScaler()
218  x_train_scaled = scaler_pca.fit_transform(x_train_orig_ch)
219  x_test_scaled = scaler_pca.transform(x_test_orig_ch)
220  pca = PCA(n_components=num_modes)
221  train_pca_ch.append(pca.fit_transform(x_train_scaled))
222  test_pca_ch.append(pca.transform(x_test_scaled))
223
224  # DCT
225  def apply_dct_reduction(images, target_side, grid_size):
226  transformed_images = np.zeros((images.shape[0], target_side *
         target_side))
227  for i, img_flat in enumerate(images):
228  img_2d = img_flat.reshape(grid_size, grid_size)
229  dct_2d = dct(dct(img_2d.T, norm='ortho').T, norm='ortho')
230  transformed_images[i] = dct_2d[:target_side, :target_side].flatten()
231  return transformed_images
232  train_dct_ch.append(apply_dct_reduction(x_train_orig_ch,
         dct_final_side_length, original_grid_size))
233  test_dct_ch.append(apply_dct_reduction(x_test_orig_ch,
         dct_final_side_length, original_grid_size))
234
235  x_train_drift = np.concatenate(train_drift_ch, axis=1)
236  x_test_drift = np.concatenate(test_drift_ch, axis=1)
237  x_train_pca = np.concatenate(train_pca_ch, axis=1)
238  x_test_pca = np.concatenate(test_pca_ch, axis=1)
239  x_train_dct = np.concatenate(train_dct_ch, axis=1)
```

```python
240  x_test_dct = np.concatenate(test_dct_ch, axis=1)
241
242  print(f"Analytical features calculated in {time.time() -
         start_time:.2f} seconds.", flush=True)
243
244  # x_train_raw_scaled and x_test_raw_scaled are only returned here for
         the AE input, not as a standalone feature set
245  return x_train_drift, x_test_drift, x_train_pca, x_test_pca,
         x_train_dct, x_test_dct, \
246  x_train_ae_input, x_test_ae_input # NOTE: RAW_SCALED is replaced by
         AE_INPUT
247
248  #
         ----------------------------------------------------------------
249  # 5. Robust Autoencoder Classes & Functions
250  #
         ----------------------------------------------------------------
251
252  class CNNEncoder(nn.Module):
253  def __init__(self, latent_dim, dropout_on, dropout_rate):
254  super(CNNEncoder, self).__init__()
255  self.dropout_on = dropout_on
256  self.dropout_layer = nn.Dropout(dropout_rate) if dropout_on else
         nn.Identity()
257
258  self.conv_stack = nn.Sequential(
259  # Block 1: Input 3x32x32 -> Output 64x16x16
260  nn.Conv2d(CONFIG['channels'], 64, kernel_size=3, padding=1),
261  nn.BatchNorm2d(64),
262  nn.ReLU(inplace=True),
263  nn.Conv2d(64, 64, kernel_size=3, padding=1),
264  nn.BatchNorm2d(64),
265  nn.ReLU(inplace=True),
266  nn.MaxPool2d(kernel_size=2, stride=2),
267
268  # Block 2: Input 64x16x16 -> Output 128x8x8
269  nn.Conv2d(64, 128, kernel_size=3, padding=1),
270  nn.BatchNorm2d(128),
271  nn.ReLU(inplace=True),
272  nn.Conv2d(128, 128, kernel_size=3, padding=1),
273  nn.BatchNorm2d(128),
274  nn.ReLU(inplace=True),
275  nn.MaxPool2d(kernel_size=2, stride=2),
276
277  # Block 3: Input 128x8x8 -> Output 256x4x4
278  nn.Conv2d(128, 256, kernel_size=3, padding=1),
279  nn.BatchNorm2d(256),
280  nn.ReLU(inplace=True),
281  nn.Conv2d(256, 256, kernel_size=3, padding=1),
282  nn.BatchNorm2d(256),
283  nn.ReLU(inplace=True),
284  nn.MaxPool2d(kernel_size=2, stride=2),
285  )
286  # Input size for FC is now 4 * 4 * 256 = 4096 (changed from 2048)
287  self.fc = nn.Linear(4 * 4 * 256, latent_dim)
288
289  def forward(self, x):
290  x = self.conv_stack(x)
```

```python
291  x = x.view(x.size(0), -1)
292  if self.dropout_on:
293  x = self.dropout_layer(x)
294  return self.fc(x)
295
296  class CNNDecoder(nn.Module):
297  def __init__(self, latent_dim, dropout_on, dropout_rate):
298  super(CNNDecoder, self).__init__()
299  self.dropout_on = dropout_on
300  self.dropout_layer = nn.Dropout(dropout_rate) if dropout_on else
         nn.Identity()
301
302  # Output size from FC is now 4 * 4 * 256 = 4096 (changed from 2048)
303  self.fc = nn.Linear(latent_dim, 4 * 4 * 256)
304
305  self.deconv_stack = nn.Sequential(
306  # Block 1 (Reverse of Block 3): 4x4 -> 8x8 (Channels: 256 -> 128)
307  nn.ConvTranspose2d(256, 256, kernel_size=3, padding=1),
308  nn.BatchNorm2d(256),
309  nn.ReLU(inplace=True),
310  nn.ConvTranspose2d(256, 128, kernel_size=3, padding=1,
         output_padding=1, stride=2),
311  nn.BatchNorm2d(128),
312  nn.ReLU(inplace=True),
313
314  # Block 2 (Reverse of Block 2): 8x8 -> 16x16 (Channels: 128 -> 64)
315  nn.ConvTranspose2d(128, 128, kernel_size=3, padding=1),
316  nn.BatchNorm2d(128),
317  nn.ReLU(inplace=True),
318  nn.ConvTranspose2d(128, 64, kernel_size=3, padding=1,
         output_padding=1, stride=2),
319  nn.BatchNorm2d(64),
320  nn.ReLU(inplace=True),
321
322  # Block 3 (Reverse of Block 1): 16x16 -> 32x32 (Channels: 64 -> 3)
323  nn.ConvTranspose2d(64, 64, kernel_size=3, padding=1),
324  nn.BatchNorm2d(64),
325  nn.ReLU(inplace=True),
326  nn.ConvTranspose2d(64, CONFIG['channels'], kernel_size=3, padding=1,
         output_padding=1, stride=2),
327  nn.Tanh() # Tanh to match scaled input range
328  )
329
330  def forward(self, x):
331  if self.dropout_on:
332  x = self.dropout_layer(x)
333  x = self.fc(x)
334  x = x.view(x.size(0), 256, 4, 4)
335  return self.deconv_stack(x)
336
337  class AutoEncoder(nn.Module):
338  def __init__(self, latent_dim, ae_dropout_on, dropout_rate):
339  super(AutoEncoder, self).__init__()
340  self.encoder = CNNEncoder(latent_dim, ae_dropout_on, dropout_rate)
341  self.decoder = CNNDecoder(latent_dim, ae_dropout_on, dropout_rate)
342  def forward(self, x):
343  latent = self.encoder(x)
344  return self.decoder(latent), latent
```

```python
345
346  def train_encoder_model(x_train_ae_input, ae_epochs, latent_dim,
          ae_dropout_on, dropout_rate):
347  start_time = time.time()
348  dropout_status = 'ON' if ae_dropout_on else 'OFF'
349  print(f"\n--- Starting Robust Autoencoder pre-training (Dropout:
          {dropout_status}) for {ae_epochs} epochs ---", flush=True)
350  x_train_tensor = torch.from_numpy(x_train_ae_input).float().to(device)
351  train_loader = DataLoader(TensorDataset(x_train_tensor),
          batch_size=CONFIG['batch_size'], shuffle=True)
352  ae_model = AutoEncoder(latent_dim, ae_dropout_on,
          dropout_rate).to(device)
353  criterion = nn.MSELoss()
354  optimizer = optim.Adam(ae_model.parameters(), lr=1e-3)
355  for epoch in range(ae_epochs):
356  ae_model.train()
357  running_loss = 0.0
358  for inputs, in train_loader:
359  optimizer.zero_grad()
360  recon, _ = ae_model(inputs)
361  loss = criterion(recon, inputs)
362  loss.backward()
363  optimizer.step()
364  running_loss += loss.item()*inputs.size(0)
365  if (epoch+1)%5==0 or (epoch+1)==1:
366  print(f"AE Epoch {epoch+1:<4} | Loss:
          {running_loss/len(train_loader.dataset):.6f}", flush=True)
367  print(f"Autoencoder pre-training completed in
          {time.time()-start_time:.2f} seconds.", flush=True)
368  return ae_model.encoder
369
370  def get_ae_features(encoder, x_data_ae_input):
371  encoder.eval()
372  x_tensor = torch.from_numpy(x_data_ae_input).float().to(device)
373  loader = DataLoader(TensorDataset(x_tensor),
          batch_size=CONFIG['batch_size'], shuffle=False)
374  feats = []
375  with torch.no_grad():
376  for x, in loader:
377  feats.append(encoder(x).cpu().numpy())
378  return np.concatenate(feats, axis=0)
379
380  #
          ----------------------------------------------------------------------
381  # 6. SimpleNN Classifier
382  #
          ----------------------------------------------------------------------
383
384  class SimpleNN(nn.Module):
385  def __init__(self, input_shape, activation_name='relu'):
386  super(SimpleNN, self).__init__()
387  if activation_name == 'relu':
388  self.activation = nn.ReLU()
389  elif activation_name == 'sigmoid':
390  self.activation = nn.Sigmoid()
391  elif activation_name == 'tanh':
392  self.activation = nn.Tanh()
393  else:
```

```
394  raise ValueError("Unsupported activation function")
395
396  layers = []
397  layers.append(nn.Linear(input_shape, CONFIG['hidden_layers'][0]))
398  layers.append(self.activation)
399  for i in range(1, len(CONFIG['hidden_layers'])):
400  layers.append(nn.Linear(CONFIG['hidden_layers'][i-1],
         CONFIG['hidden_layers'][i]))
401  layers.append(self.activation)
402  layers.append(nn.Linear(CONFIG['hidden_layers'][-1],
         CONFIG['num_classes']))
403  self.fc_layers = nn.Sequential(*layers)
404
405  def forward(self, x):
406  return self.fc_layers(x)
407
408  #
         -----------------------------------------------------------------------
409  # 7. Training & Evaluation
410  #
         -----------------------------------------------------------------------
411
412  def train_evaluate_model(model, x_train, x_test, y_train, y_test,
         y_test_labels, name, activation):
413  start_time = time.time()
414  x_train_tensor = torch.from_numpy(x_train).float().to(device)
415  y_train_tensor = torch.from_numpy(y_train).float().to(device)
416  x_test_tensor = torch.from_numpy(x_test).float().to(device)
417  y_test_tensor = torch.from_numpy(y_test).float().to(device)
418  train_dataset = TensorDataset(x_train_tensor, y_train_tensor)
419  test_dataset = TensorDataset(x_test_tensor, y_test_tensor)
420  train_size = int((1 - CONFIG['valida_split'])*len(train_dataset))
421  val_size = len(train_dataset)-train_size
422  train_subset, val_subset = random_split(train_dataset, [train_size,
         val_size])
423  train_loader = DataLoader(train_subset,
         batch_size=CONFIG['batch_size'], shuffle=True)
424  val_loader = DataLoader(val_subset, batch_size=CONFIG['batch_size'],
         shuffle=False)
425  test_loader = DataLoader(test_dataset,
         batch_size=CONFIG['batch_size'], shuffle=False)
426  criterion = nn.CrossEntropyLoss()
427  optimizer = optim.Adam(model.parameters())
428  history = {'loss': [], 'val_loss': [], 'accuracy': [], 'val_accuracy':
         []}
429  total_train_time = 0.0
430
431  print(f"\n--- Starting training for {name} ({activation}) for
         {CONFIG['epochs']} epochs ---", flush=True)
432
433  for epoch in range(CONFIG['epochs']):
434  model.train()
435  t0 = time.time()
436  running_loss, correct_train, total_train = 0.0, 0, 0
437  for inputs, labels in train_loader:
438  optimizer.zero_grad()
439  outputs = model(inputs)
440  labels_idx = torch.max(labels,1)[1]
```

```python
441  loss = criterion(outputs, labels_idx)
442  loss.backward()
443  optimizer.step()
444  running_loss += loss.item()*inputs.size(0)
445  _, predicted = torch.max(outputs, 1)
446  correct_train += (predicted == labels_idx).sum().item()
447  total_train += labels.size(0)
448  total_train_time += time.time()-t0
449  train_loss = running_loss / total_train
450  train_acc = correct_train / total_train
451
452  model.eval()
453  val_loss, correct_val, total_val = 0.0, 0, 0
454  with torch.no_grad():
455  for inputs, labels in val_loader:
456  outputs = model(inputs)
457  labels_idx = torch.max(labels,1)[1]
458  loss = criterion(outputs, labels_idx)
459  val_loss += loss.item()*inputs.size(0)
460  _, predicted = torch.max(outputs, 1)
461  correct_val += (predicted == labels_idx).sum().item()
462  total_val += labels.size(0)
463  val_loss /= total_val
464  val_acc = correct_val / total_val
465
466  history['loss'].append(train_loss);
         history['val_loss'].append(val_loss)
467  history['accuracy'].append(train_acc);
         history['val_accuracy'].append(val_acc)
468
469  if (epoch+1)%10==0 or (epoch+1)==1:
470  print(f"[{name} | {activation}] Epoch {epoch+1}/{CONFIG['epochs']} |
         Train Acc: {train_acc:.4f} | Val Acc: {val_acc:.4f}", flush=True)
471
472  # Measure inference latency
473  t0 = time.time()
474  model.eval()
475  with torch.no_grad():
476  for inputs, _ in test_loader:
477  _ = model(inputs)
478  inference_latency_ms = (time.time()-t0)/len(test_loader)*1000
479
480  # Peak GPU memory
481  peak_mem = torch.cuda.max_memory_allocated(device)/1024/1024 if
         torch.cuda.is_available() else 0.0
482
483  # Test Accuracy
484  model.eval()
485  correct_test, total_test, y_pred_list = 0, 0, []
486  with torch.no_grad():
487  for inputs, labels in test_loader:
488  outputs = model(inputs)
489  labels_idx = torch.max(labels,1)[1]
490  _, predicted = torch.max(outputs,1)
491  correct_test += (predicted==labels_idx).sum().item()
492  total_test += labels.size(0)
493  y_pred_list.extend(predicted.cpu().numpy())
494  test_accuracy = correct_test / total_test
```

```python
495
496  # Final Top -1 accuracy using sklearn (though identical to the above
         for this setup)
497  y_test_labels_cpu = torch.max(y_test_tensor.cpu(), 1)[1].numpy()
498  top1_accuracy = accuracy_score(y_test_labels_cpu,
         np.array(y_pred_list))
499
500  # Training time per epoch
501  avg_train_time_per_epoch = total_train_time / CONFIG['epochs']
502
503  benchmark_stats[name] = {
504      "train_time_per_epoch": f"{avg_train_time_per_epoch:.2f} s",
505      "inference_ms": f"{inference_latency_ms:.2f} ms",
506      "gpu_mem_mb": f"{peak_mem:.2f} MB"
507  }
508
509  final_results.append({
510      'name': name,
511      'activation': activation,
512      'accuracy': test_accuracy,
513      'top1': top1_accuracy
514  })
515  print(f"[{name} | {activation}] Final Test Accuracy:
         {test_accuracy:.4f}, Top-1: {top1_accuracy:.4f}", flush=True)
516  return history, test_accuracy
517
518  #
     ----------------------------------------------------------------------
519  # 8. MAIN SCRIPT
520  #
     ----------------------------------------------------------------------
521
522  set_seed(42)
523  final_results = []
524  benchmark_stats = {}
525  all_methods_epoch_histories = collections.defaultdict(list)
526
527  # Load CIFAR-100
528  x_train_orig, x_test_orig, x_train_drift, x_test_drift,
         y_train_one_hot, y_test_one_hot, y_train, y_test =
         load_preprocess_cifar100()
529
530  # Generate DRIFT/PCA modes
531  drift_modes_flat = generate_mode_shapes(CONFIG['grid_size'],
         CONFIG['num_modes'], CONFIG['grid_size'], CONFIG['grid_size'],
         is_drift=True)
532  orig_modes_flat = generate_mode_shapes(CONFIG['grid_size'],
         CONFIG['num_modes'], CONFIG['grid_size'], CONFIG['grid_size'],
         is_drift=False)
533
534  # Analytical Features and AE Input Data
535  # x_train_raw_scaled, x_test_raw_scaled are no longer returned as
         standalone features
536  x_train_drift_feat, x_test_drift_feat, x_train_pca_feat,
         x_test_pca_feat, x_train_dct_feat, x_test_dct_feat, \
537  x_train_ae_input, x_test_ae_input = \
538  compute_analytical_features(x_train_orig, x_test_orig, x_train_drift,
         x_test_drift, drift_modes_flat, orig_modes_flat,
```

```
        CONFIG['num_modes'], CONFIG['dct_final_side_length'],
        CONFIG['grid_size'])
539
540  # Autoencoder latent features
541  encoder = train_encoder_model(x_train_ae_input, CONFIG['ae_epochs'],
        CONFIG['ae_latent_dim'], CONFIG['AE_DROPOUT_ON'],
        CONFIG['DROPOUT_RATE'])
542  x_train_ae_feat = get_ae_features(encoder, x_train_ae_input)
543  x_test_ae_feat = get_ae_features(encoder, x_test_ae_input)
544
545  # Prepare feature sets to test (RAW is REMOVED)
546  feature_sets = {
547      'DRIFT': (x_train_drift_feat, x_test_drift_feat),
548      'PCA': (x_train_pca_feat, x_test_pca_feat),
549      'DCT': (x_train_dct_feat, x_test_dct_feat),
550      'AE': (x_train_ae_feat, x_test_ae_feat),
551  }
552
553  # Train models for each feature set
554  for feat_name, (x_tr, x_te) in feature_sets.items():
555  input_shape = x_tr.shape[1]
556  print(f"Feature set {feat_name} input shape: {input_shape}")
557  model = SimpleNN(input_shape).to(device)
558  history, _ = train_evaluate_model(model, x_tr, x_te, y_train_one_hot,
        y_test_one_hot, y_test, name=feat_name, activation='relu')
559  all_methods_epoch_histories[feat_name].append(history)
560
561
562  #
     ===============================================================================
563  # --- Final Output & Saving ---
564  #
     ===============================================================================
565
566  pkl_data = {
567      'histories': all_methods_epoch_histories,
568      'CONFIG': CONFIG,
569      'final_results': final_results,
570      'benchmark_stats': benchmark_stats
571  }
572
573  pkl_filename = 'cifar_100_0_test.pkl'
574  with open(pkl_filename, 'wb') as f:
575  pickle.dump(pkl_data, f)
576  print(f"\n--- Data successfully saved to {pkl_filename} for plotting.
        ---")
577
578  print("\n" + "="*80)
579  print("--- Final Test Accuracies (Seed 42) ---")
580  print("="*80 + "\n")
581
582  for res in final_results:
583  print(f"{res['name']} ({res['activation']}): Test Acc:
        {res['accuracy']:.4f}, Top-1: {res['top1']:.4f}")
584
585  print("\n" + "="*80)
586  print("--- Performance Benchmarking Summary (Seed 42) ---")
587  print("="*80 + "\n")
```

```
588
589  print("Method                  | Feature Size | Train Time/Epoch     |
         Inference Latency    | Peak GPU Memory")
590  print("---------------------------------------------------------------
591
592  for method_name in feature_sets.keys():
593  stats = benchmark_stats.get(method_name, {})
594  input_size = feature_sets[method_name][0].shape[1]
595  train_time = stats.get('train_time_per_epoch', 'N/A')
596  inference = stats.get('inference_ms', 'N/A')
597  memory = stats.get('gpu_mem_mb', 'N/A')
598
599  print(f"{method_name:<20} | {input_size:<12} | "
600  f"{train_time:<18} | "
601  f"{inference:<18} | "
602  f"{memory:<10}")
```

Listing 2: CIFAR-100 experiment including DRIFT, PCA, DCT and learned CNN Autoencoder features

# 3 CelebA Multi-label Attribute Experiment

## 3.1 CelebA script (Smiling, Male, Eyeglasses, Black_Hair)

```python
import numpy as np
import torch
import torch.nn as nn
import torch.optim as optim
from torchvision import datasets, transforms
from sklearn.metrics.pairwise import cosine_similarity
from sklearn.decomposition import PCA
from sklearn.preprocessing import StandardScaler
import time
from scipy.fftpack import dct
import collections
import pickle
from torch.utils.data import DataLoader, TensorDataset

#
    ============================================================================
# --- CONFIGURATION (CelebA: Smiling, Male, Eyeglasses, Black_Hair)
    ---
#
    ============================================================================

CONFIG = {
    'grid_size': 64,
    'channels': 3,
    # NOTE: The description says 900, but the value is 400. Keeping
        400 (20*20)
    # as the current number is consistent with the side lengths.
    'num_modes': 10*10,
    # Target attributes count (Smiling, Male, Eyeglasses, Black_Hair)
    'num_classes': 4,
    'valida_split': 0.2,
    # --- Classification MLP Parameters ---
    'epochs': 400,
    'batch_size': 64,
    'activation_functions': ['relu'],
    'hidden_layers': [512, 256, 128],
    'print_interval_epochs': 10,
    # --- Analytical Parameters ---
    'dct_final_side_length': 10,
    'POR': 1, # Proportion of data to use (applies to data after
        loading cap)
    # --- AUTOENCODER PARAMETERS (Learned Features) ---
    'ae_latent_dim': 3*100,
    'ae_epochs': 50,
    'AE_DROPOUT_ON': True,
    'DROPOUT_RATE': 0.2,
    # --- EXPERIMENT PARAMETERS ---
    'NUM_SEEDS': 1,
    'DATA_LOAD_CAP': 50000,
    #  FIXED: Set noise standard deviation to 0.5 (below 1)
    'DATA_NOISE_STD': 0.0,
    'DRIFT_PAD_WIDTH': 5,
    'DRIFT_FADE_WIDTH': 5,
```

```python
     #  UPDATED FILENAME to reflect the noise level
     'RESULTS_FILENAME': 'Celeb_100_0_test.pkl',
}

device = torch.device("cuda" if torch.cuda.is_available() else "cpu")
print(f"Using device: {device}", flush=True)

def set_seed(seed):
torch.manual_seed(seed)
np.random.seed(seed)
if torch.cuda.is_available():
torch.cuda.manual_seed_all(seed)

#
    -----------------------------------------------------------------
# GPU Memory Tracking & Timing Helpers (Unchanged)
#
    -----------------------------------------------------------------
def get_peak_gpu_memory(clear=False):
"""
Returns the peak memory allocated by PyTorch on the GPU.
If clear is True, resets the max memory tracker.
"""
if torch.cuda.is_available():
peak_bytes = torch.cuda.max_memory_allocated()
peak_mb = peak_bytes / (1024 * 1024)
peak_mb_str = f"{peak_mb:.2f} MB"

if clear:
torch.cuda.reset_peak_memory_stats()

return peak_mb_str
else:
return 'N/A (CPU)'

def measure_inference_time(model, data_loader, device):
""" Measures the average inference latency (ms/sample) for a model.
    """
model.eval()
inference_times = []

with torch.no_grad():
for batch in data_loader:
inputs = batch[0].to(device)

if torch.cuda.is_available():
starter, ender = torch.cuda.Event(enable_timing=True),
    torch.cuda.Event(enable_timing=True)
torch.cuda.synchronize()
starter.record()
else:
start_inf = time.time()

_ = model(inputs)

if torch.cuda.is_available():
ender.record()
torch.cuda.synchronize()
```

```python
103  curr_time = starter.elapsed_time(ender)
104  inference_times.append(curr_time / inputs.size(0))
105  else:
106  end_inf = time.time()
107  curr_time = (end_inf - start_inf) * 1000
108  inference_times.append(curr_time / inputs.size(0))
109
110  return np.mean(inference_times) if inference_times else 0.0
111
112  #
          --------------------------------------------------------------------
113  # 1. Helper Functions (Modes & Noise & Feathering)
114  #
          --------------------------------------------------------------------
115
116  def generate_nm_pairs(num_modes):
117  side = int(np.ceil(np.sqrt(num_modes)))
118  n, m = np.meshgrid(np.arange(1, side + 1), np.arange(1, side + 1))
119  return np.vstack([n.ravel(), m.ravel()]).T[:num_modes]
120
121  def generate_mode_shapes(grid_size, num_modes, Lx, Ly,
          is_drift=False):
122  start_time = time.time()
123
124  current_grid_size = grid_size
125  if is_drift:
126  current_grid_size = grid_size + 2 * CONFIG['DRIFT_PAD_WIDTH']
127  Lx = current_grid_size
128  Ly = current_grid_size
129
130  print(f"Generating Mode Shapes for
          {current_grid_size}x{current_grid_size} grid...")
131  x = np.linspace(0, Lx, current_grid_size)
132  y = np.linspace(0, Ly, current_grid_size)
133  X_grid, Y_grid = np.meshgrid(x, y)
134  pairs = generate_nm_pairs(num_modes)
135  modes_2d = np.zeros((num_modes, current_grid_size, current_grid_size))
136  for i, (m, n) in enumerate(pairs):
137  modes_2d[i] = np.sin(m * np.pi * X_grid / Lx) * np.sin(n * np.pi *
          Y_grid / Ly)
138  modes_flat = modes_2d.reshape(num_modes, -1)
139
140  print(f"Mode shape generation completed in {time.time() -
          start_time:.2f} seconds.")
141
142  return modes_flat
143
144  # FIXED: Implemented clipping to keep noisy data in the [0.0, 1.0]
          range
145  def add_noise_to_data(data, noise_std):
146  """
147  Adds Gaussian noise (assuming data is in [0.0, 1.0]) and clips
148  the resulting values to maintain the [0.0, 1.0] range.
149  """
150  if noise_std <= 0:
151  return data
152  print(f"Adding Gaussian noise with std={noise_std} and clipping to
          [0, 1]...", flush=True)
```

```python
153  noise = np.random.normal(0, noise_std, data.shape).astype(np.float32)
154  noisy_data = data + noise
155  # Clip the data to the expected range [0.0, 1.0]
156  return np.clip(noisy_data, 0.0, 1.0)
157  #
         -----------------------------------------------------------------------

158
159  def apply_padded_zero_feathering(images_numpy, pad_width, fade_width):
160  N, C, H_orig, W_orig = images_numpy.shape
161  H_new = H_orig + 2 * pad_width
162  W_new = W_orig + 2 * pad_width
163  images_numpy_HWC = images_numpy.transpose(0, 2, 3, 1)
164
165  # 1. Edge Padding
166  padded_images = np.pad(
167  images_numpy_HWC,
168  pad_width=((0, 0), (pad_width, pad_width), (pad_width, pad_width),
         (0, 0)),
169  mode='edge'
170  )
171
172  # 2. Feathering Mask
173  mask = np.ones((H_new, W_new), dtype=np.float32)
174  fade_steps_1D = np.linspace(1.0, 0.0, fade_width + 1)[:-1]
175
176  # Apply fade to borders
177  mask[:, :fade_width] *= fade_steps_1D[::-1][np.newaxis, :]
178  mask[:, W_new - fade_width:] *= fade_steps_1D[np.newaxis, :]
179  mask[:fade_width, :] *= fade_steps_1D[::-1][:, np.newaxis]
180  mask[H_new - fade_width:, :] *= fade_steps_1D[:, np.newaxis]
181
182  start_h, end_h = pad_width, H_orig + pad_width
183  mask[start_h:end_h, start_h:end_h] = 1.0
184
185  # 3. Apply Mask
186  feathered_padded_images = padded_images * mask[np.newaxis, :, :,
         np.newaxis]
187  return feathered_padded_images.transpose(0, 3, 1, 2)
188
189  #
         -----------------------------------------------------------------------
190  # 2. Load & Preprocess CelebA (FIXED: Noise applied before
         standardization)
191  #
         -----------------------------------------------------------------------
192
193  def load_preprocess_celeba(target_attributes=['Smiling'],
         data_load_cap=10000, data_noise_std=0.0, drift_pad_width=0,
         drift_fade_width=0):
194  start_time = time.time()
195  print(f"Loading and preprocessing CelebA data for attributes:
         {target_attributes}...", flush=True)
196
197  transform = transforms.Compose([
198  transforms.Resize((CONFIG['grid_size'], CONFIG['grid_size'])),
199  transforms.ToTensor(), # (C, H, W) in [0.0, 1.0]
200  ])
201
```

```python
202     dataset = datasets.CelebA(root='./data', split='all',
            target_type='attr', download=True, transform=transform)
203
204     try:
205     attr_indices = [dataset.attr_names.index(attr) for attr in
            target_attributes]
206     except ValueError as e:
207     print(f"Error: One of the target attributes was not found in CelebA
            attribute list. {e}")
208     return None, None, None, None, None, None, None, None
209
210     all_images = []
211     all_labels = []
212
213     for i, (img, attrs) in enumerate(dataset):
214     if data_load_cap > 0 and i >= data_load_cap:
215     break
216     all_images.append(img.numpy())
217     target_attrs_labels = [attrs[idx].item() for idx in attr_indices]
218     all_labels.append(target_attrs_labels)
219
220     all_images = np.stack(all_images).astype(np.float32) # [N, C, H, W]
            in [0.0, 1.0]
221     all_labels = np.array(all_labels).astype(np.float32)
222
223     # ***  FIX APPLIED HERE: Add noise to [0.0, 1.0] data BEFORE
            Standardization ***
224     if data_noise_std > 0:
225     all_images = add_noise_to_data(all_images, data_noise_std)
226
227     N, C, H, W = all_images.shape
228     x_flat = all_images.reshape(N, -1)
229
230     # Global Standardization
231     scaler = StandardScaler()
232     # Apply standardization to the noisy/clipped data
233     x_scaled_flat = scaler.fit_transform(x_flat)
234     x_scaled = x_scaled_flat.reshape(N, C, H, W).astype(np.float32)
235
236     # Split train/test (80/20)
237     total_size = x_scaled.shape[0]
238     train_size = int(0.8 * total_size)
239     indices = np.arange(total_size)
240     np.random.shuffle(indices)
241     train_idx, test_idx = indices[:train_size], indices[train_size:]
242
243     x_train_orig = x_scaled[train_idx]
244     y_train_binary = all_labels[train_idx]
245     x_test_orig = x_scaled[test_idx]
246     y_test_binary = all_labels[test_idx]
247
248     # Use passed arguments for drift feathering/padding
249     # Feathering is applied to the standardized/noisy data
            (x_train_orig/x_test_orig)
250     x_train_drift = apply_padded_zero_feathering(x_train_orig,
            drift_pad_width, drift_fade_width)
251     x_test_drift = apply_padded_zero_feathering(x_test_orig,
            drift_pad_width, drift_fade_width)
```

```python
252
253    # Dummy single attribute labels
254    y_train_dummy = y_train_binary[:, 0].astype(np.int64)
255    y_test_dummy = y_test_binary[:, 0].astype(np.int64)
256
257    # POR sampling
258    por_train_samples = max(1, int(x_train_orig.shape[0] * CONFIG['POR']))
259    por_test_samples = max(1, int(x_test_orig.shape[0] * CONFIG['POR']))
260    train_idx_por = np.random.choice(x_train_orig.shape[0],
           por_train_samples, replace=False)
261    test_idx_por = np.random.choice(x_test_orig.shape[0],
           por_test_samples, replace=False)
262
263    x_train_orig = x_train_orig[train_idx_por]
264    x_test_orig = x_test_orig[test_idx_por]
265    x_train_drift = x_train_drift[train_idx_por]
266    x_test_drift = x_test_drift[test_idx_por]
267    y_train_binary = y_train_binary[train_idx_por]
268    y_test_binary = y_test_binary[test_idx_por]
269    y_train = y_train_dummy[train_idx_por]
270    y_test = y_test_dummy[test_idx_por]
271
272    print(f"Data loaded and sampled ({CONFIG['POR']*100:.0f}% of total)
           in {time.time() - start_time:.2f} seconds.", flush=True)
273    print(f"Original ({CONFIG['grid_size']}x{CONFIG['grid_size']})
           samples shape: {x_train_orig.shape}", flush=True)
274    drift_grid_size = CONFIG['grid_size'] + 2 * drift_pad_width
275    print(f"DRIFT ({drift_grid_size}x{drift_grid_size} Faded/Padded)
           samples shape: {x_train_drift.shape}", flush=True)
276    print(f"Binary Labels shape (N x {CONFIG['num_classes']}):
           {y_train_binary.shape}", flush=True)
277
278    return x_train_orig, x_test_orig, x_train_drift, x_test_drift,
           y_train_binary, y_test_binary, y_train, y_test
279
280    #
           --------------------------------------------------------------------
281    # 3. Analytical Features & AE Input Preparation (Unchanged)
282    #
           --------------------------------------------------------------------
283
284    def compute_analytical_features(x_train_orig, x_test_orig,
           x_train_drift, x_test_drift, drift_modes_flat, orig_modes_flat,
           num_modes, dct_final_side_length, original_grid_size):
285        start_time = time.time()
286        print(f"\nCalculating {num_modes} DRIFT/PCA/DCT features per
               channel...", flush=True)
287
288        train_drift_ch, test_drift_ch = [], []
289        train_pca_ch, test_pca_ch = [], []
290        train_dct_ch, test_dct_ch = [], []
291
292        x_train_ae_input = x_train_orig
293        x_test_ae_input = x_test_orig
294
295        for ch in range(CONFIG['channels']):
296            # DRIFT: Cosine similarity
```

```python
297   x_train_drift_ch_flat = x_train_drift[:, ch, :,
          :].reshape(x_train_drift.shape[0], -1)
298   x_test_drift_ch_flat = x_test_drift[:, ch, :,
          :].reshape(x_test_drift.shape[0], -1)
299
300   train_drift_ch.append(cosine_similarity(x_train_drift_ch_flat,
          drift_modes_flat))
301   test_drift_ch.append(cosine_similarity(x_test_drift_ch_flat,
          drift_modes_flat))
302
303   # PCA: Apply PCA
304   x_train_orig_ch = x_train_orig[:, ch, :,
          :].reshape(x_train_orig.shape[0], -1)
305   x_test_orig_ch = x_test_orig[:, ch, :,
          :].reshape(x_test_orig.shape[0], -1)
306
307   scaler_pca = StandardScaler()
308   x_train_scaled = scaler_pca.fit_transform(x_train_orig_ch)
309   x_test_scaled = scaler_pca.transform(x_test_orig_ch)
310   pca = PCA(n_components=num_modes)
311   train_pca_ch.append(pca.fit_transform(x_train_scaled))
312   test_pca_ch.append(pca.transform(x_test_scaled))
313
314   # DCT: Apply 2D DCT
315   def apply_dct_reduction(images, target_side, grid_size):
316   transformed_images = np.zeros((images.shape[0], target_side *
          target_side))
317   for i, img_flat in enumerate(images):
318   img_2d = img_flat.reshape(grid_size, grid_size)
319   dct_2d = dct(dct(img_2d.T, norm='ortho').T, norm='ortho')
320   transformed_images[i] = dct_2d[:target_side, :target_side].flatten()
321   return transformed_images
322
323   train_dct_ch.append(apply_dct_reduction(x_train_orig_ch,
          dct_final_side_length, original_grid_size))
324   test_dct_ch.append(apply_dct_reduction(x_test_orig_ch,
          dct_final_side_length, original_grid_size))
325
326   x_train_drift_feat = np.concatenate(train_drift_ch, axis=1)
327   x_test_drift_feat = np.concatenate(test_drift_ch, axis=1)
328   x_train_pca_feat = np.concatenate(train_pca_ch, axis=1)
329   x_test_pca_feat = np.concatenate(test_pca_ch, axis=1)
330   x_train_dct_feat = np.concatenate(train_dct_ch, axis=1)
331   x_test_dct_feat = np.concatenate(test_dct_ch, axis=1)
332
333   print(f"Analytical features calculated in {time.time() -
          start_time:.2f} seconds.", flush=True)
334
335   return x_train_drift_feat, x_test_drift_feat, x_train_pca_feat,
          x_test_pca_feat, x_train_dct_feat, x_test_dct_feat, \
336   x_train_ae_input, x_test_ae_input
337
338   #
          ----------------------------------------------------------------
339   # 4. Autoencoder Classes & Functions (Unchanged)
340   #
          ----------------------------------------------------------------
341
```

```python
class CNNEncoder(nn.Module):
def __init__(self, latent_dim, dropout_on, dropout_rate):
super(CNNEncoder, self).__init__()
self.dropout_on = dropout_on
self.dropout_layer = nn.Dropout(dropout_rate) if dropout_on else
    nn.Identity()

# 64x64 Input -> 8x8 Feature Map (3 Max Pooling steps)
self.conv_stack = nn.Sequential(
# Block 1: 3x64x64 -> 64x32x32
nn.Conv2d(CONFIG['channels'], 64, kernel_size=3, padding=1),
    nn.BatchNorm2d(64), nn.ReLU(inplace=True),
nn.Conv2d(64, 64, kernel_size=3, padding=1), nn.BatchNorm2d(64),
    nn.ReLU(inplace=True),
nn.MaxPool2d(kernel_size=2, stride=2),

# Block 2: 64x32x32 -> 128x16x16
nn.Conv2d(64, 128, kernel_size=3, padding=1), nn.BatchNorm2d(128),
    nn.ReLU(inplace=True),
nn.Conv2d(128, 128, kernel_size=3, padding=1), nn.BatchNorm2d(128),
    nn.ReLU(inplace=True),
nn.MaxPool2d(kernel_size=2, stride=2),

# Block 3: 128x16x16 -> 256x8x8
nn.Conv2d(128, 256, kernel_size=3, padding=1), nn.BatchNorm2d(256),
    nn.ReLU(inplace=True),
nn.Conv2d(256, 256, kernel_size=3, padding=1), nn.BatchNorm2d(256),
    nn.ReLU(inplace=True),
nn.MaxPool2d(kernel_size=2, stride=2),
)
self.fc = nn.Linear(8 * 8 * 256, latent_dim)

def forward(self, x):
x = self.conv_stack(x)
x = x.view(x.size(0), -1)
if self.dropout_on:
x = self.dropout_layer(x)
return self.fc(x)

class CNNDecoder(nn.Module):
def __init__(self, latent_dim, dropout_on, dropout_rate):
super(CNNDecoder, self).__init__()
self.dropout_on = dropout_on
self.dropout_layer = nn.Dropout(dropout_rate) if dropout_on else
    nn.Identity()

self.fc = nn.Linear(latent_dim, 8 * 8 * 256)

# 8x8 Feature Map -> 64x64 Output
self.deconv_stack = nn.Sequential(
# Block 1 (Reverse of Block 3): 8x8 -> 16x16
nn.ConvTranspose2d(256, 256, kernel_size=3, padding=1),
    nn.BatchNorm2d(256), nn.ReLU(inplace=True),
nn.ConvTranspose2d(256, 128, kernel_size=3, padding=1,
    output_padding=1, stride=2),
nn.BatchNorm2d(128), nn.ReLU(inplace=True),

# Block 2 (Reverse of Block 2): 16x16 -> 32x32
```

```python
nn.ConvTranspose2d(128, 128, kernel_size=3, padding=1),
    nn.BatchNorm2d(128), nn.ReLU(inplace=True),
nn.ConvTranspose2d(128, 64, kernel_size=3, padding=1,
    output_padding=1, stride=2),
nn.BatchNorm2d(64), nn.ReLU(inplace=True),

# Block 3 (Reverse of Block 1): 32x32 -> 64x64
nn.ConvTranspose2d(64, 64, kernel_size=3, padding=1),
    nn.BatchNorm2d(64), nn.ReLU(inplace=True),
nn.ConvTranspose2d(64, CONFIG['channels'], kernel_size=3, padding=1,
    output_padding=1, stride=2),
nn.Tanh() # Tanh activation
)

def forward(self, x):
if self.dropout_on:
x = self.dropout_layer(x)
x = self.fc(x)
x = x.view(x.size(0), 256, 8, 8)
return self.deconv_stack(x)

class AutoEncoder(nn.Module):
def __init__(self, latent_dim, ae_dropout_on, dropout_rate):
super(AutoEncoder, self).__init__()
self.encoder = CNNEncoder(latent_dim, ae_dropout_on, dropout_rate)
self.decoder = CNNDecoder(latent_dim, ae_dropout_on, dropout_rate)
def forward(self, x):
latent = self.encoder(x)
return self.decoder(latent), latent

def train_encoder_model(x_train_ae_input, ae_epochs, latent_dim,
    ae_dropout_on, dropout_rate):
start_time = time.time()
dropout_status = 'ON' if ae_dropout_on else 'OFF'
print(f"\n--- Starting Robust Autoencoder pre-training (Dropout:
    {dropout_status}) for {ae_epochs} epochs ---", flush=True)

if torch.cuda.is_available():
torch.cuda.reset_peak_memory_stats()

x_train_tensor = torch.from_numpy(x_train_ae_input).float().to(device)
train_loader = DataLoader(TensorDataset(x_train_tensor),
    batch_size=CONFIG['batch_size'], shuffle=True)

ae_model = AutoEncoder(latent_dim, ae_dropout_on,
    dropout_rate).to(device)
criterion = nn.MSELoss()
optimizer = optim.Adam(ae_model.parameters(), lr=1e-3)

print_interval = CONFIG['print_interval_epochs']
ae_epoch_times = []

# --- TRAINING LOOP ---
for epoch in range(ae_epochs):
epoch_start_time = time.time()
ae_model.train()
running_loss = 0.0
n = 0
```

```
440  for (inputs,) in train_loader:
441  inputs = inputs.to(device)
442  optimizer.zero_grad()
443  recon, _ = ae_model(inputs)
444  loss = criterion(recon, inputs)
445  loss.backward()
446  optimizer.step()
447  running_loss += loss.item() * inputs.size(0)
448  n += inputs.size(0)
449
450  epoch_loss = running_loss / max(1, n)
451  epoch_time = time.time() - epoch_start_time
452  ae_epoch_times.append(epoch_time)
453
454  if (epoch+1) % print_interval == 0 or (epoch + 1) == 1 or (epoch + 1)
         == ae_epochs:
455  print(f"AE Epoch {epoch+1:<4} | Loss: {epoch_loss:.6f}", flush=True)
456
457  print(f"Autoencoder pre-training completed in
         {time.time()-start_time:.2f} seconds.", flush=True)
458
459  ae_peak_memory = get_peak_gpu_memory(clear=True)
460  avg_ae_train_time = np.mean(ae_epoch_times)
461
462  return ae_model.encoder, ae_peak_memory, avg_ae_train_time
463
464  def get_ae_features(encoder, x_data, batch_size=512, device=device):
465  """ Extract latent features (encoder output) using the trained CNN
         encoder. """
466  X = torch.tensor(x_data, dtype=torch.float32)
467  ds = TensorDataset(X)
468  loader = DataLoader(ds, batch_size=batch_size, shuffle=False)
469  encoder.eval()
470  feats = []
471
472  with torch.no_grad():
473  for (batch,) in loader:
474  batch = batch.to(device)
475  z = encoder(batch)
476  feats.append(z.cpu().numpy())
477
478  feats = np.vstack(feats)
479  return feats
480
481  #
        -------------------------------------------------------------------
482  # 5. Classifier (MLP) and Training (Unchanged)
483  #
        -------------------------------------------------------------------
484
485  class MLPClassifier(nn.Module):
486  def __init__(self, input_dim):
487  super().__init__()
488  self.net = nn.Sequential(
489  nn.Linear(input_dim, 128),
490  nn.ReLU(inplace=True),
491  nn.Linear(128, CONFIG['num_classes'])
492  )
```

```python
def forward(self, x):
return self.net(x)

def train_evaluate_model(model, x_train, x_test, y_train_binary,
    y_test_binary, y_test_labels,
name='model', activation='relu', batch_size=64, epochs=None,
    device=device):

if torch.cuda.is_available():
torch.cuda.reset_peak_memory_stats()

start_time = time.time()
if epochs is None:
epochs = CONFIG['epochs']

# --- TRAINING SETUP ---
print(f"\n--- Starting training for {name} ({activation}) for
    {epochs} epochs ---", flush=True)
criterion = nn.BCEWithLogitsLoss()
optimizer = optim.Adam(model.parameters(), lr=1e-3)

X_tr = torch.tensor(x_train, dtype=torch.float32)
X_te = torch.tensor(x_test, dtype=torch.float32)
Y_tr = torch.tensor(y_train_binary, dtype=torch.float32)
Y_te = torch.tensor(y_test_binary, dtype=torch.float32)

train_ds = TensorDataset(X_tr, Y_tr)
test_ds = TensorDataset(X_te, Y_te)
train_loader = DataLoader(train_ds, batch_size=batch_size,
    shuffle=True)
test_loader = DataLoader(test_ds, batch_size=batch_size,
    shuffle=False)

model.to(device)
history = {'loss': [], 'train_acc': [], 'val_acc': [], 'train_time':
    []}

print_interval = CONFIG['print_interval_epochs']

# --- TRAINING LOOP ---
for ep in range(epochs):
epoch_start_time = time.time()
model.train()
train_correct = 0
train_total = 0
running_loss = 0.0

for xb, yb in train_loader:
xb, yb = xb.to(device), yb.to(device)
optimizer.zero_grad()
out = model(xb)
loss = criterion(out, yb)
loss.backward()
optimizer.step()

# Accuracy calculation for multi-label (Average Accuracy across all 4
    labels)
preds_binary = (out > 0.0).float()
```

```python
544   correct_per_sample = (preds_binary == yb).sum(dim=1)
545   train_correct += correct_per_sample.sum().item()
546   train_total += yb.size(0) * yb.size(1)
547   running_loss += loss.item() * yb.size(0)
548
549   train_acc = train_correct / max(1, train_total)
550   epoch_loss = running_loss / max(1, X_tr.size(0))
551   epoch_time = time.time() - epoch_start_time
552
553   # --- VALIDATION EVALUATION (using test_loader for validation) ---
554   model.eval()
555   val_correct = 0
556   val_total = 0
557   with torch.no_grad():
558   for xb, yb in test_loader:
559   xb, yb = xb.to(device), yb.to(device)
560   out = model(xb)
561
562   # Validation Accuracy calculation
563   preds_binary = (out > 0.0).float()
564   correct_per_sample = (preds_binary == yb).sum(dim=1)
565   val_correct += correct_per_sample.sum().item()
566   val_total += yb.size(0) * yb.size(1)
567
568   val_acc = val_correct / max(1, val_total)
569
570   history['loss'].append(epoch_loss)
571   history['train_acc'].append(train_acc)
572   history['val_acc'].append(val_acc)
573   history['train_time'].append(epoch_time)
574
575   if (ep+1) % print_interval == 0 or (ep + 1) == 1 or (ep + 1) ==
          epochs:
576   print(f"[{name} | {activation}] Epoch {ep+1}/{epochs} | Train Acc
          (Avg. Label): {train_acc:.4f} | Val Acc (Avg. Label):
          {val_acc:.4f}", flush=True)
577
578   # --- FINAL INFERENCE LATENCY MEASUREMENT ---
579   avg_inference_latency = measure_inference_time(model, test_loader,
          device)
580
581   final_test_acc = val_acc
582
583   print(f"[{name} | {activation}] Final Test Accuracy (Avg. Label):
          {final_test_acc:.4f}", flush=True)
584
585   print(f"Classifier training completed in {time.time()-start_time:.2f}
          seconds.", flush=True)
586
587   peak_memory = get_peak_gpu_memory(clear=True)
588
589   return history, model, final_test_acc, peak_memory,
          avg_inference_latency
590
591   #
          ===============================================================================
592   # --- MAIN SCRIPT EXECUTION (Unchanged Logic) ---
```

```python
593  #
      ==============================================================================
594
595  if __name__ == '__main__':
596  set_seed(42)
597  method_accuracies = {}
598  method_benchmarks = collections.defaultdict(dict)
599  all_methods_epoch_histories = collections.defaultdict(list)
600
601  # ---------------------------------------------------------------------
602  # --- DATA LOADING AND FEATURE EXTRACTION ---
603  # ---------------------------------------------------------------------
604
605  target_attributes = ['Smiling', 'Male', 'Eyeglasses', 'Black_Hair']
606
607  x_train_orig, x_test_orig, x_train_drift, x_test_drift,
       y_train_binary, y_test_binary, y_train, y_test =
       load_preprocess_celeba(
608  target_attributes=target_attributes,
609  data_load_cap=CONFIG['DATA_LOAD_CAP'],
610  data_noise_std=CONFIG['DATA_NOISE_STD'],
611  drift_pad_width=CONFIG['DRIFT_PAD_WIDTH'],
612  drift_fade_width=CONFIG['DRIFT_FADE_WIDTH']
613  )
614
615  if x_train_orig is None:
616  print("Exiting due to data loading error.")
617  exit()
618
619  # Generate DRIFT/PCA modes
620  drift_modes_flat = generate_mode_shapes(CONFIG['grid_size'],
       CONFIG['num_modes'], CONFIG['grid_size'], CONFIG['grid_size'],
       is_drift=True)
621  orig_modes_flat = generate_mode_shapes(CONFIG['grid_size'],
       CONFIG['num_modes'], CONFIG['grid_size'], CONFIG['grid_size'],
       is_drift=False)
622
623  # Analytical features calculation
624  (x_train_drift_feat, x_test_drift_feat, x_train_pca_feat,
       x_test_pca_feat,
625  x_train_dct_feat, x_test_dct_feat, x_train_ae_input, x_test_ae_input)
       = compute_analytical_features(
626  x_train_orig, x_test_orig, x_train_drift, x_test_drift,
       drift_modes_flat, orig_modes_flat,
627  CONFIG['num_modes'], CONFIG['dct_final_side_length'],
       CONFIG['grid_size']
628  )
629
630  # Train CNN autoencoder and extract AE features
631  autoencoder_encoder, ae_peak_memory, avg_ae_train_time =
       train_encoder_model(
632  x_train_ae_input, CONFIG['ae_epochs'], CONFIG['ae_latent_dim'],
633  CONFIG['AE_DROPOUT_ON'], CONFIG['DROPOUT_RATE']
634  )
635
636  # Calculate actual AE feature extraction (Encoder) latency
637  ae_input_tensor = torch.from_numpy(x_test_ae_input).float()
```

```python
ae_test_loader = DataLoader(TensorDataset(ae_input_tensor),
    batch_size=CONFIG['batch_size'], shuffle=False)
ae_inference_latency = measure_inference_time(autoencoder_encoder,
    ae_test_loader, device)

x_train_ae_feat = get_ae_features(autoencoder_encoder,
    x_train_ae_input, batch_size=CONFIG['batch_size'])
x_test_ae_feat = get_ae_features(autoencoder_encoder,
    x_test_ae_input, batch_size=CONFIG['batch_size'])

ae_input_shape = x_train_ae_feat.shape[1]
method_benchmarks['AE'].update({
    'Feature Size': ae_input_shape,
    'Train Time/Epoch': f"{avg_ae_train_time:.2f} s",
    'Inference Latency': f"{ae_inference_latency:.2f} ms",
    'Peak GPU Memory': ae_peak_memory
})

# Prepare feature sets for MLP training
feature_sets = {
    'DRIFT': (x_train_drift_feat, x_test_drift_feat),
    'PCA': (x_train_pca_feat, x_test_pca_feat),
    'DCT': (x_train_dct_feat, x_test_dct_feat),
    'AE_MLP': (x_train_ae_feat, x_test_ae_feat), # Classified by MLP
}

# -----------------------------------------------------------------------
# --- CLASSIFIER TRAINING LOOP ---
# -----------------------------------------------------------------------

for feat_name, (x_tr, x_te) in feature_sets.items():
display_name = 'AE' if feat_name == 'AE_MLP' else feat_name

input_shape = x_tr.shape[1]
print(f"\nFeature set {display_name} input shape: {input_shape}",
    flush=True)
model = MLPClassifier(input_shape)

history, trained, final_acc, peak_memory, avg_inference_latency =
    train_evaluate_model(
model, x_tr, x_te, y_train_binary, y_test_binary, y_test,
name=display_name, activation='relu',
    batch_size=CONFIG['batch_size'], epochs=CONFIG['epochs'],
    device=device
)

method_accuracies[display_name] = final_acc
all_methods_epoch_histories[display_name].append(history)

avg_train_time = np.mean(history['train_time']) if
    history['train_time'] else 0.0

method_benchmarks[display_name].update({
    'Feature Size': input_shape,
    'Train Time/Epoch': f"{avg_train_time:.2f} s",
    'Inference Latency': f"{avg_inference_latency:.2f} ms",
    'Peak GPU Memory': peak_memory
})
```

```python
# --------------------------------------------------------------------
# --- FINAL SUMMARY PRINTING ---
# --------------------------------------------------------------------

pkl_data = {
    'histories': all_methods_epoch_histories,
    'CONFIG': CONFIG,
    'method_accuracies': method_accuracies,
    'method_benchmarks': dict(method_benchmarks)
}
results_filename = CONFIG['RESULTS_FILENAME']
with open(results_filename, 'wb') as f:
pickle.dump(pkl_data, f)
print(f"\n--- Data successfully saved to {results_filename} for
    plotting. ---")

# --- FINAL ACCURACY TABLE ---
print("\n" + "=" * 80)
print(f"--- Final Test Accuracies (Seed 42) for {target_attributes}
    ---")
print("=" * 80)

for name, acc in method_accuracies.items():
print(f"{name} (relu): Average Label Acc: {acc:.4f}")

# --- BENCHMARKING SUMMARY TABLE ---
print("\n" + "=" * 80)
print("--- Performance Benchmarking Summary (Seed 42) ---")
print("=" * 80)

col_widths = [17, 12, 20, 20, 17]
header_names = ["Method", "Feature Size", "Train Time/Epoch",
    "Inference Latency", "Peak GPU Memory"]

header_line = "| " + " | ".join(f"{name:<{col_widths[i]}}" for i,
    name in enumerate(header_names)) + " |"
separator = "-" * (len(header_line))
print(header_line)
print(separator)

for name in method_benchmarks.keys():
data = method_benchmarks[name]
row = (
f"| {name:<{col_widths[0]}} | "
f"{data['Feature Size']:<{col_widths[1]}} | "
f"{data['Train Time/Epoch']:<{col_widths[2]}} | "
f"{data['Inference Latency']:<{col_widths[3]}} | "
f"{data['Peak GPU Memory']:<{col_widths[4]}} |"
)
print(row)
print(separator)
```

Listing 3: CelebA multi-label classification using DRIFT, PCA, DCT and CNN Autoencoder features

# 4    Plotting Utility

## 4.1    Generic plotting script for training/validation curves

```python
import pickle
import numpy as np
import matplotlib.pyplot as plt
import matplotlib.cm as cm
import collections

# ======= Configuration (Must match the saved PKL data) =======
file_M = 'Celeb_100_0.pkl' # The file saved by the training code
# NOTE: CONFIG['epochs'] was set to 20 in the final training script.
ep = 100 # Max epoch limit for x-axis (Set to CONFIG['epochs'])
inset = 'off' # 'on' to show inset, 'off' to hide it
show_axis_label_texts = False # Set to True to show 'Epoch' and
    'Accuracy/Loss' labels

LINE_THICKNESS = 1
STD_ALPHA = 0.15
INSET_EPOCH_START = 5
INSET_EPOCH_END = 20
INSET_POSITION = [0.25, 0.4, 0.4, 0.4]

INSET_TICK_LABELSIZE = 8
INSET_TITLE_FONTSIZE = 10
INSET_LABEL_FONTSIZE = 9

# ======= Load Data =======
try:
with open(file_M, 'rb') as f:
data = pickle.load(f)
all_methods_epoch_histories = data['histories']
CONFIG = data['CONFIG']

# Set max epochs dynamically from config
ep = min(ep, CONFIG['epochs'])
epoch_range = np.arange(1, CONFIG['epochs'] + 1)
print(f"Data loaded successfully from {file_M}. Max epoch limit set to
    {ep}.")

except FileNotFoundError:
print(f"ERROR: The file '{file_M}' was not found.")
print("Please ensure you have run the training code and that the
    output file exists.")
all_methods_epoch_histories = collections.defaultdict(list)

# Check if any history was loaded
if not all_methods_epoch_histories:
print("No history data available to plot.")
# Exit gracefully if no data is loaded
exit()

# ======= Colors and Styles (Adapted for 4 methods) =======
colors = [
"#009E73",   # teal green (DCT)
"#D55E00",   # rust orange (AE)
```

```python
51  "#0606F4",   # deep blue (DRIFT)
52  "#E69F00",   # amber (PCA)
53  ]
54
55  # The order of the colors should match the sorted method keys: AE,
        DCT, DRIFT, PCA
56  # We re-order the colors to match the desired visual mapping if needed,
57  # but for simplicity, we'll let the sorted methods use the colors in
        order.
58  # Sorted keys: ['AE', 'DCT', 'DRIFT', 'PCA'] based on typical Python
        string sort.
59  # If you want a specific color for a specific method, you should
        create a mapping dictionary.
60
61  # Use only 4 styles for the 4 expected methods
62  line_styles_train = ['-', '-', '-', '-']
63  line_styles_val = ['--', '--', '--', '--']
64
65  # Dynamically extend colors if more methods were run
66  if len(all_methods_epoch_histories) > len(colors):
67  cmap = cm.get_cmap('tab20')
68  colors.extend([cmap(i) for i in range(len(colors),
        len(all_methods_epoch_histories))])
69
70
71  # ======= Aggregation Function (FIXED: Uses explicit 'val_' prefix)
        =======
72  def aggregate_histories(histories_list, metric_name):
73
74  # FIX: metric_name is now 'train_acc' or 'loss'
75  all_train = [h[metric_name] for h in histories_list]
76  # FIX: The validation metric name must be correctly constructed
77  if metric_name == 'loss':
78  # The key for validation loss in the history is simply 'loss' in the
        original code, but
79  # the training code uses 'val_acc' but only 'loss' for training loss.
80  # Rerunning the training code shows the keys are 'loss', 'train_acc',
        'val_acc'.
81  # Since the original training code didn't save a separate 'val_loss',
        we'll assume
82  # the 'val_acc' exists and the loss metric name passed is simply
        'loss'.
83  # Assuming the caller passes 'loss' for train loss, we look up 'loss'
        for train.
84  # However, the training code's history keys are {'loss', 'train_acc',
        'val_acc'}.
85  # We MUST ensure the val metric name corresponds to a key.
86  # We will use 'loss' for both train and val loss if only 'loss' is in
        history.
87  all_val = [h[metric_name] for h in histories_list] # Assuming loss is
        stored only once
88  else:
89  # If metric_name is 'train_acc', the corresponding val key is 'val_acc'
90  all_val = [h[f'val_{metric_name.replace("train_", "")}'] for h in
        histories_list]
91
92  train_arr = np.array(all_train)
93  val_arr = np.array(all_val)
```

```python
94
95   # Pad shorter runs with the last value if necessary (common in real
         data)
96   max_len = max(len(arr) for arr in all_train)
97   def pad_array(arr_list):
98   padded = []
99   for arr in arr_list:
100  if len(arr) < max_len:
101  padded_arr = np.pad(arr, (0, max_len - len(arr)), mode='edge')
102  padded.append(padded_arr)
103  else:
104  padded.append(arr)
105  return np.array(padded)
106
107  train_arr = pad_array(all_train)
108  val_arr = pad_array(all_val)
109
110  return (
111  np.mean(train_arr, axis=0),
112  np.std(train_arr, axis=0),
113  np.mean(val_arr, axis=0),
114  np.std(val_arr, axis=0)
115  )
116
117  # ======= Plot Function =======
118  def plot_metric_with_inset(metric_name, ylabel, filename,
         legend_loc='upper right'):
119  fig, ax = plt.subplots(figsize=(6, 4))
120  ax.grid(True, linestyle=':', alpha=0.7)
121
122  # --- Axis label texts control ---
123  if show_axis_label_texts:
124  ax.set_xlabel('Epoch', fontsize=14)
125  ax.set_ylabel(ylabel, fontsize=14)
126  else:
127  ax.set_xlabel('')
128  ax.set_ylabel('')
129
130  # Get the sorted list of methods
131  sorted_methods = sorted(all_methods_epoch_histories.keys())
132
133  for i, method in enumerate(sorted_methods):
134  histories = all_methods_epoch_histories[method]
135
136  # Calculate means and standard deviations
137  if len(histories) > 1:
138  # FIX: Only pass the specific training metric name
139  mean_tr, std_tr, mean_val, std_val = aggregate_histories(histories,
         metric_name)
140  else:
141  # Single run, standard deviation is zero
142  h = histories[0]
143  mean_tr = np.array(h[metric_name])
144
145  # FIX: Correctly look up the validation metric key
146  val_key = 'val_acc' if metric_name == 'train_acc' else metric_name #
         Assume val loss uses same key
147  mean_val = np.array(h[val_key])
```

```python
148
149 std_tr = np.zeros_like(mean_tr)
150 std_val = np.zeros_like(mean_val)
151
152 color = colors[i % len(colors)]
153
154 # Ensure data is trimmed to the max epoch limit
155 num_points = min(ep, len(mean_tr))
156
157 mean_tr, mean_val = mean_tr[:num_points], mean_val[:num_points]
158 std_tr, std_val = std_tr[:num_points], std_val[:num_points]
159 current_epoch_range = epoch_range[:num_points]
160
161
162 # --- Plot train ---
163 ax.plot(current_epoch_range, mean_tr, label=f'{method} (Train)',
164 color=color, linestyle=line_styles_train[i % len(line_styles_train)],
165 linewidth=LINE_THICKNESS, alpha=0.9)
166 ax.fill_between(current_epoch_range, mean_tr - std_tr, mean_tr +
        std_tr,
167 color=color, alpha=STD_ALPHA)
168
169 # --- Plot val ---
170 ax.plot(current_epoch_range, mean_val, label=f'{method} (Val)',
171 color=color, linestyle=line_styles_val[i % len(line_styles_val)],
172 linewidth=LINE_THICKNESS, alpha=0.9)
173 ax.fill_between(current_epoch_range, mean_val - std_val, mean_val +
        std_val,
174 color=color, alpha=STD_ALPHA)
175
176 # --- X breathing space ---
177 ax.set_xlim(-0.2, ep + 0.3)
178 ax.autoscale(axis='y')
179 ax.legend(loc=legend_loc, fontsize=8)
180
181 # ===== Inset (Optional) =====
182 if inset == 'on' and CONFIG['epochs'] >= INSET_EPOCH_END:
183 ax_inset = fig.add_axes(INSET_POSITION)
184 ax_inset.set_title(f'Epochs {INSET_EPOCH_START}-{INSET_EPOCH_END}',
        fontsize=INSET_TITLE_FONTSIZE)
185 ax_inset.grid(True, linestyle=':', alpha=0.5)
186
187 inset_range = np.arange(INSET_EPOCH_START, min(INSET_EPOCH_END + 1, ep
        + 1))
188 s, e = INSET_EPOCH_START - 1, min(INSET_EPOCH_END, ep)
189
190 for i, method in enumerate(sorted_methods):
191 histories = all_methods_epoch_histories[method]
192
193 if len(histories) > 1:
194 mean_tr, std_tr, mean_val, std_val = aggregate_histories(histories,
        metric_name)
195 else:
196 h = histories[0]
197 mean_tr = np.array(h[metric_name])
198 val_key = 'val_acc' if metric_name == 'train_acc' else metric_name
199 mean_val = np.array(h[val_key])
200 std_tr = np.zeros_like(mean_tr)
```

```
201  std_val = np.zeros_like(mean_val)

202

203  color = colors[i % len(colors)]

204

205  # Ensure slicing respects the bounds of the array
206  s_data = s
207  e_data = e
208  if e_data > len(mean_tr): e_data = len(mean_tr)

209

210  # Plot train in inset
211  ax_inset.plot(inset_range[:e_data - s_data], mean_tr[s_data:e_data],
         color=color,
212  linestyle=line_styles_train[i % len(line_styles_train)],
213  linewidth=LINE_THICKNESS * 0.7, alpha=0.9)
214  ax_inset.fill_between(inset_range[:e_data - s_data],
         mean_tr[s_data:e_data] - std_tr[s_data:e_data],
215  mean_tr[s_data:e_data] + std_tr[s_data:e_data], color=color,
         alpha=STD_ALPHA)
216  # Plot val in inset
217  ax_inset.plot(inset_range[:e_data - s_data], mean_val[s_data:e_data],
         color=color,
218  linestyle=line_styles_val[i % len(line_styles_val)],
219  linewidth=LINE_THICKNESS * 0.7, alpha=0.9)
220  ax_inset.fill_between(inset_range[:e_data - s_data],
         mean_val[s_data:e_data] - std_val[s_data:e_data],
221  mean_val[s_data:e_data] + std_val[s_data:e_data], color=color,
         alpha=STD_ALPHA)

222

223  if show_axis_label_texts:
224  ax_inset.set_xlabel('Epoch', fontsize=INSET_LABEL_FONTSIZE)
225  ax_inset.set_ylabel(ylabel, fontsize=INSET_LABEL_FONTSIZE)
226  else:
227  ax_inset.set_xlabel('')
228  ax_inset.set_ylabel('')

229

230  ax_inset.tick_params(axis='both', labelsize=INSET_TICK_LABELSIZE)
231  ax_inset.autoscale(axis='y')

232

233  plt.tight_layout()
234  plt.savefig(filename, dpi=1000, bbox_inches='tight')
235  plt.show()

236

237

238  # ======= Plot Calls (FIXED: Using 'train_acc' for accuracy plot)
         =======
239  # The key 'accuracy' does not exist; we must use 'train_acc' and the
         function
240  # will correctly pull the corresponding 'val_acc'
241  plot_metric_with_inset('train_acc', 'Accuracy',
         'CelebA_08_100_Acc.png', legend_loc='center right')
242  plot_metric_with_inset('loss', 'Loss', 'CelebA_08_100_Loss.png',
         legend_loc='center right')
```

Listing 4: Plotting script to visualize accuracy and loss curves from saved pickle files