# OpenReview forum: "DRIFT: DATA REDUCTION VIA INFORMATIVE FEATURE TRANSFORMATION – GENERALIZATION BEGINS BEFORE DEEP LEARNING STARTS"
_ICLR.cc/2026/Conference — Submitted to ICLR 2026_

### Official Review · Reviewer_BdYW · 2025-10-27

**Soundness:** 2
**Presentation:** 1
**Contribution:** 2
**Rating:** 2
**Confidence:** 3

**Summary:**

This paper aims to capture some essential data patterns in real data via DRIFT. DRIFT, a novel preprocessing method, leverages vibrational mode analysis to extract resonant features from input data, enhancing training stability and generalization.  The experiments in this paper show that DRIFT outperforms PCA and DCT on MNIST and CIFAR100 in dimensionality reduction, convergence behavior, and overfitting mitigation.

**Strengths:**

1. This paper provide a novel physics-inspired perspective, which is different with previous PCA and DCT.

2. This method shows well performance and robustness in MNIST and CIFAR100.

**Weaknesses:**

1. The presentation is poor. Authors should spend more time to polish their introduction.

2. The text in the graph is difficult to read.

3. The empirical evidence is limited. Authors should provide results on more datasets.

**Questions:**

How do the physical validity and sensitivity impact the performance?

---

> ### Author Response · Authors · 2025-11-27
>
> We do appreciate the time and attention from the reviewer:
> We have updated the manuscript addressing following notes (Q is for questions and W for weaknesses that are not covered in the Questions):
>
> Presentation (W1):
> -
> The authors confirmed they have polished the text and improved the clarity of the introduction and explanations throughout the paper.
>
> Visual Clarity (W2):
> -
> All figures and graphs now have clearer labels and legends to improve readability.
>
> Empirical Scope (W3):
> -
> The experiments were expanded to include the CelebA dataset in addition to MNIST and CIFAR-100. The method's robustness under noise was evaluated, and updated training and test accuracy plots were provided.
>
> Physical Impact (Q1):
> -
> The physical modeling choice is deliberate and sound:
> Images date pixels are treated as deformations of a plate. We use simply supported boundary conditions (pixels = 0 at edges, linearly changing near edges), which is physically meaningful and allows for a closed-form modal decomposition.
> The approach is guided by plate theory, where the lower-energy modes dominate the overall response. By focusing the feature extraction on these dominant low-energy modes, the method ensures that the resulting DRIFT features are both physically grounded and effective in practice (e.g., extracting salient patterns and filtering noise). More detailed explanations provided in the Model section as well as the order of cost analysis and more complex dataset experiments.

---

### Official Review · Reviewer_2my8 · 2025-10-27

**Soundness:** 1
**Presentation:** 1
**Contribution:** 1
**Rating:** 2
**Confidence:** 4

**Summary:**

The main idea of the paper, as I understand it, is to derive physics-inspired lower-dimensional features of natural images, and use these features as input for more robust image classification. The physics-inspired features come from modeling the image as a physical object, calculating its vibrational modes, and projecting the image onto the top set of vibration modes to create a low-dimensional feature representation. Then this representation is validated for image reconstruction (to assess how much of the semantic content is captured by the lower-dimensional features) and classification, where use of the proposed features makes classification models more robust to hyperparameters (learning rate) and less prone to overfitting.

**Strengths:**

The idea is novel to my knowledge; I haven’t seen other papers that interpret an image as an elastic plate and use vibrational modes to extract features. The extracted features do capture some of the image content and seem to mostly work for image classification, with less overfitting and more robustness to learning rate choice compared to the selected baselines. In my view the robustness to learning rates is the strongest result presented.

**Weaknesses:**

Conceptual weakness:

The central idea of using vibrational modes to extract image features is not well motivated. It is presented as physics-inspired, but the connection to physics is tenuous and based on the seemingly arbitrary decision to interpret the grayscale value in an image as the height of an imagined elastic plate.

Experimental weaknesses:

There are 2 types of experiments, showing image reconstruction from limited features/modes (on MNIST), and showing image classification accuracy and loss (for both MNIST and CIFAR-100). These datasets are fairly simple tasks, and the results show at best marginal improvement over baselines (and often no improvement).
- In the MNIST reconstruction quality experiment, PCA (low-rank features) produces higher quality reconstructions than the proposed DRIFT features, both quantitatively and visually.
- In the MNIST classification experiments, in some settings classification from raw pixels is best, in some cases DRIFT is best, and in some cases DCT features are best. Even when DRIFT is best on test data, the improvement is marginal (maybe half a percentage point better than PCA or raw pixels, and about the same as DCT).
- In the CIFAR-100 classification experiments, the improvement in smaller generalization gap is achieved more by lowering the train accuracy than by raising the test accuracy.

Presentation weaknesses:

- The formatting of the paper does not match the ICLR template. The text is wider (left and right margins are smaller) than the template, for both the abstract and the main text.
- The text has section breaks, but no paragraph breaks. The text within each section is a single continuous block.
- The introduction section has the related work discussion as part (or really the majority) of it. This is unusual, though not necessarily an issue if it could be internally structured into paragraphs that group related works together and explain how they relate to the proposed method.
- The figures are not formatted for ease of interpretation. Figure 3 on the right reports standard deviation, but does not specify what this is computed over (ie how many images). It also doesn’t specify what dataset is used for Figure 3, though from context I think this is MNIST. - - The text within the figures is quite small, legible only by zooming in, which is insufficient for readers who may choose to print a paper. This is especially true for figures 5, 6, and 7, which are completely uninterpretable without zooming in.
- Much of the text in the results section is devoted to rote description of the results in the figures. This is unnecessary; it would be more valuable to devote this space to a short but clear interpretation of the results, and to providing some motivation for the main idea in the introduction.
- The bottom of page 7 lists advantages of the proposed DRIFT features. The first and third bullet points are redundant with each other; both describe robustness to high learning rates.

**Questions:**

See weaknesses; if any of these can be clarified that would be helpful.

---

> ### Author Response · Authors · 2025-11-27
>
> We do appreciate the time and attention from the reviewer:
> We have updated the manuscript addressing following notes (Q is for questions and W for weaknesses that are not covered in the Questions):
>
> Presentation(W1,W2,W3,W4):
> -
>
> We acknowledge that some sections of the paper were dense and the narrative flow could be improved. In the revision, we have restructured the main exposition to provide a clearer, coherent thread connecting generalization theory and the motivation for DRIFT. Dense explanations are simplified, and illustrative examples are added where appropriate, to make the paper easier to follow while maintaining technical rigor. Also made sure the template format is met.
> Additionally, the introduction is now framed at a more conceptual level, focusing on the broader ideas behind the mode.
> Figures are being reproduced for better visibility even for the print purpose and better explanation provided.
>
> Conceptual Motivation (W):
> -
> The analogy is purposeful and grounded in established physical theory. DRIFT treats an image as a 2D field whose pixel intensities correspond to the deformation of a simply supported elastic plate, allowing us to leverage well-studied vibration modes. The simply supported boundary conditions (values fixed at edges with smooth linear decay) map directly to physical constraints, ensuring a meaningful and well-behaved modal decomposition rather than an arbitrary basis choice. By projecting onto the dominant low-energy vibrational modes, the modes requiring the least energy to excite, DRIFT captures the most salient and stable variations in the image while maintaining smoothness and reducing sensitivity to noise. This physics-inspired representation yields a closed-form, analytically tractable basis that improves both efficiency and robustness, as confirmed in our experiments on MNIST, CIFAR-100, and CelebA (including noisy settings). Additional details and cost analyses are now included in the Model section.
>
> Experimental Weakness (W):
> -
> The goal of DRIFT is not to maximize early reconstruction quality or achieve the steepest initial training curve, but to extract meaningful, generalizable features that enhance stability and robustness. This design choice naturally leads to a gentler training trajectory: the model avoids overfitting to redundant or highly specific patterns and instead learns representations that transfer better. Importantly, this does not imply slower convergence, our reported training time per epoch and inference latency show that DRIFT remains computationally comparable to other model baselines. In practice, this behavior yields improved generalization, as reflected in the more stable test performance across MNIST, CIFAR-100, and CelebA, especially under noisy conditions. Overall, the results indicate that DRIFT’s features reduce overfitting  which stands out as one of the method’s strongest advantages. Expanded experiments on CelebA and noisy settings further substantiate these findings.

---

### Official Review · Reviewer_X9VA · 2025-10-28

**Soundness:** 1
**Presentation:** 1
**Contribution:** 1
**Rating:** 0
**Confidence:** 4

**Summary:**

The paper proposes a physically inspired method for dimensionality reduction in image data. The approach draws on vibration-mode analysis from structural mechanics and represents each image as a linear combination of spatial wave patterns (mode shapes) of increasing complexity. This reparametrisation serves as a preprocessing step for feature compression, analogous to PCA and DCT, which are used as baselines. Experiments on MNIST and CIFAR-100 aim to demonstrate that networks trained on the proposed features achieve smoother convergence, greater stability under high learning rates, and improved generalisation.

The general idea of exploring efficient, learning-free, low-dimensional image representations to stabilise training is interesting and conceptually appealing. However, the paper provides insufficient theoretical justification or empirical evidence for the claimed advantages. The exposition is overly dense and lacks a clear motivating thread, the thin-plate analogy is weakly grounded for complex imagery, and the experiments remain limited to toy datasets and shallow architectures. Strengthening the motivation for where the proposed representation is most relevant, extending experiments to more realistic image domains and architectures, and situating the work within the broader literature on image encoders and feature learning would substantially improve the paper.

**Strengths:**

- **S1: Physics-inspired image representations.**
The paper explores alternative low-dimensional image representations by introducing a physics-inspired basis derived from vibration-mode analysis. The general effort to move beyond standard encodings such as PCA or DCT is commendable. More broadly, incorporating physics-based principles into deep learning remains a worthwhile and original direction.

- **S2: Potential for improved learning stability.**
The preliminary results showing greater stability under high learning rates suggest some promise in the proposed representation. The idea of enhancing learning robustness and training dynamics through a simple, low-dimensional image encoding is interesting and could have practical relevance in domains where stability and interpretability are critical, such as medical image analysis (e.g., CT or X-ray imagery), where image structures are often smooth and may be better approximated by the thin-plate model assumed in the method. Extending the same principles to 3D data could also prove fruitful.

**Weaknesses:**

- **W1: Paper structure.** The paper is generally poorly written and difficult to follow. The introduction and overall exposition are overly dense and insufficiently structured. The narrative moves abruptly from generalisation theory to dimensionality reduction and finally to the proposed method without a coherent motivating thread.

- **W2: Conceptual coherence and intuition.**
The rationale for introducing vibration-mode analysis as a solution to the image representation problem is not convincingly developed. While the central intuition may hold for simple imagery such as MNIST, it remains unclear what kind of “fundamental physical patterns” in highly complex, real-world images could be captured by the proposed thin-plate deformation model. The generalisation of this paradigm beyond small or smooth, low-frequency images is therefore highly questionable.

- **W3: Weak experimental validation and lack of connection to modern image representation work.** All experiments are conducted on toy datasets (MNIST, CIFAR-100) and shallow MLP architectures, limiting the practical value of the results. The paper provides no evidence of scalability to modern convolutional or transformer-based architectures, nor any comparison to learning-based encoders. Research on learning low-dimensional image manifolds is vast and mature ([1-3]), yet remains unacknowledged. Furthermore, even within the presented experiments, the dimensionality-reduction results in Figure 3 show no consistent advantage over PCA in reconstruction accuracy. Combined with the lack of clear theoretical motivation, the overall contribution remains ambiguous: what concrete problem does the proposed method solve in practice?


References

[1] Radford, A., Kim, J. W., Hallacy, C., Ramesh, A., Goh, G., Agarwal, S., Sastry, G., Askell, A., Mishkin, P., Clark, J., Krueger, G., & Sutskever, I. (2021). Learning Transferable Visual Models from Natural Language Supervision (CLIP). NeurIPS 2021.

[2] Caron, M., Touvron, H., Misra, I., Jégou, H., Mairal, J., Bojanowski, P., & Joulin, A. (2021). Emerging Properties in Self-Supervised Vision Transformers (DINO). arXiv:2104.14294.

[3] He, K., Chen, X., Xie, S., Li, Y., Dollár, P., & Girshick, R. (2022). Masked Autoencoders Are Scalable Vision Learners (MAE). arXiv:2111.06377.

**Questions:**

The main questions are already reflected in the weaknesses section, but to highlight a few key points:

 - What do the authors consider the **principal result** of the paper that demonstrates its **practical relevance**? In other words, which specific contributions should the work be evaluated on at a top-tier level?

- How do the authors envision **scaling** the proposed method to **real-world, complex imagery** and modern architectures such as CNNs or Vision Transformers?

- How does the proposed approach compare in relevance and potential to established, powerful **image encoders** such as CLIP?

- What kind of **useful implicit bias** does the proposed representation introduce (relative to DCT or PCA) when applied to complex, high-dimensional images?

---

> ### Author Response · Authors · 2025-11-27
>
> We do appreciate the time and attention from the reviewer:
> We have updated the manuscript addressing following notes (Q is for questions and W for weaknesses that are not covered in the Questions):
>
> Paper Structure (W1):
> -
> We acknowledge that some sections of the paper were dense and the narrative flow could be improved. In the revision, we have restructured the main exposition to provide a clearer, coherent thread connecting generalization theory and the motivation for DRIFT. Dense explanations are simplified, and clearer illustrative examples are added where appropriate, to make the paper easier to follow while maintaining technical rigor.
>
> Conceptual Coherence (W2):
> -
> DRIFT models images to prioritize low-energy, smooth patterns, capturing meaningful global and mid-level features that generalize to complex images like CelebA. The closed-form basis reduces redundancy and improves generalization. Stability check is confirmed through noisy data as well as comparison to convolutional based autoencoders
>
> Weak Experimental Validation (W3):
> -
> Added experiments on CelebA and compared results to Autoencoder (AE)-based encoders. Reconstruction-focused experiments were removed to improve focus on generalization and stability. Also as brought more detail in the Model section, we have verified the order of costs while emphasizing that main goal of DRIFT is to maintain robust convergence and generalization gap reduction.
>
> We have reviewed and added references and discussed their relation to low-dimensional manifold learning, clarifying how DRIFT complements learning-based encoders by providing a physically grounded, closed-form basis that emphasizes meaningful, low-energy features.
>
> Practical Relevance (Q1,Q3):
> -
> The principal contribution is: improved generalization and stable learning (on MNIST, CIFAR-100, and CelebA), robustness to noise, and effective dimensionality reduction using a computationally efficient, interpretable basis.
>
> The proposed analytic approach (DRIFT) and established image encoders like CLIP are fundamentally relevant to different use cases. DRIFT's potential lies in its inherent interpretability and its superior noise resilience in capacity-limited tasks, as demonstrated in our CelebA experiments where its fixed, sinusoidal basis allows for stable optimization and robust feature extraction with zero feature pre-training cost. Conversely, CLIP's relevance and tremendous potential stem from its zero-shot generalization capability, achieved through massive-scale, multimodal training (vision-language alignment), enabling it to perform arbitrary visual recognition tasks on unseen data, albeit at the expense of massive computational cost and feature opacity.
>
> Scaling (Q2):
> -
> DRIFT is architecture-agnostic and acts as a front-end feature extractor. It scales naturally to higher-dimensional images (CelebA) due to its closed-form modal decomposition.
>
> Implicit Bias (Q4):
> -
> DRIFT introduces a bias toward smooth, low-energy, and physically meaningful modes, which reduces sensitivity to noise and encourages stable learning by limiting overfitting to high-frequency details.
>
> 1.	Prioritizes low-frequency, low-energy patterns that dominate the visual content, reducing sensitivity to noise and redundant features.
> 2.	Encourages stable learning by producing smoother training curves and limiting overfitting to high-frequency or dataset-specific details.
> 3.	Maintains interpretability/explainability , as each mode corresponds to a physically motivated deformation, providing insight into which structures drive the representation.
> This implicit bias allows DRIFT to provide robust, generalizable features even in complex, high-dimensional images, as demonstrated on CelebA, MNIST, and CIFAR-100, while remaining computationally efficient and architecture-agnostic.

---

### Official Review · Reviewer_xjep · 2025-10-31

**Soundness:** 3
**Presentation:** 3
**Contribution:** 3
**Rating:** 6
**Confidence:** 4

**Summary:**

This paper proposes DRIFT, a physics-inspired preprocessing that treats an image as the static deflection of a simply supported elastic plate and projects it onto low-frequency vibration mode shapes to obtain compact features. Concretely, the authors derive separable sine modes from the Kirchhoff–Love plate equation under simply supported boundary conditions and use cosine similarity between an image and each mode to produce coefficients; the first $K$ modes define a low-dimensional representation that is fixed, orthogonal, and interpretable. The method is evaluated as a drop-in front end for MLP classifiers on MNIST and CIFAR100 at various feature budgets (e.g., 16/36/49/100), and is compared with raw pixels, PCA, and DCT. Empirically, DRIFT tends to yield smoother optimization, stronger early learning, smaller generalization gaps, and reduced sensitivity to learning-rate choices; for reconstruction experiments, PCA attains the lowest mean error while DRIFT exhibits the lowest variance across mode counts and retains recognizable structure at moderate $K$. Overall, the claim is that “generalization begins before deep learning starts”: by imposing a physically motivated, low-frequency basis before training, networks can train more stably and generalize as well as or better than common linear transforms at significantly reduced input dimension.

**Strengths:**

A clear strength is the simple, interpretable construction: closed-form plate modes provide an orthogonal, low-frequency basis that is easy to compute, hardware friendly, and task agnostic; this yields stable training curves, smaller generalization gaps at small feature budgets, and robustness to aggressive learning rates, all without learning an encoder. The comparisons against PCA/DCT across multiple feature sizes and learning rates are thorough for the chosen MLP backbone, and the reconstruction analysis fairly shows DRIFT’s low variance even when PCA wins on mean MSE.

**Weaknesses:**

The algorithm relies on a specific physical idealization (rectangular grid, simply supported edges, separable sines); it is not known whether it is sensitive to the choice of boundary conditions, padding/cropping, or different shapes or resolutions of the input image. Empirical studies are limited (MNIST, CIFAR100, MLPs), and do not examine comparison to more advanced learned front ends (scattering, conv autoencoders) or convolutional backbones. Per-sample cosine features plus a norm estimate also introduce scale choices in downstream classifiers that affect task accuracy, but are not investigated. Finally, it does not relate the plate modes to the discriminative structure beyond the very general low-frequency prior, leaving open the question of when DRIFT should beat PCA/DCT.

**Questions:**

1. How sensitive are DRIFT features to the choice of simply supported edges, zero-padding, or small translations/rotations of the image? Could alternative boundry conditions or mode warping make the representation more translation/rotation tolerant without learning?

2. For RGB inputs, are modes applied per channel independently, or is there a principled 3D (space–channel) coupling?

3. How robust are DRIFT features to common corruptions such as noise or blur?

---

> ### Author Response · Authors · 2025-11-27
>
> We do appreciate the time and attention from the reviewer:
> We have updated the manuscript addressing following notes (Q is for questions and W for weaknesses that are not covered in the Questions):
>
> Theory/Sensitivity (Q1):
> -
> The choice of simply supported boundaries is a deliberate design decision for closed-form, efficient representation. New experiments under noise perturbations show stable performance, indicating the method is not overly sensitive to small variations near the boundary. Alternative boundary conditions forfeit the closed-form structure. More in depth explanation provided in the Model section. Different parameters are also added in the numerical experiment to support the efficiency of the model compared to similar concept based models.
>
> Color Input (Q2):
> -
> The modal decomposition is applied independently to each RGB channel, and the resulting feature vectors are concatenated. This was chosen for simplicity and computational efficiency.
>
> Robustness (Q3):
> -
> Expanded experiments to include controlled noise corruptions across all three datasets (MNIST, CIFAR-100, and CelebA). Results show DRIFT maintains stable and reliable performance, with smaller accuracy degradation compared to other methods. Also a comparison to a convolutional based Autoencoder provided for better clarity.

---

### Official Review · Reviewer_Q2Fw · 2025-11-01

**Soundness:** 4
**Presentation:** 3
**Contribution:** 3
**Rating:** 6
**Confidence:** 3

**Summary:**

This paper introduces DRIFT, a novel dimensionality reduction (DR) technique inspired by vibrational modes of elastic plates, projecting images onto mode shapes via cosine similarity. Evaluated on MNIST and CIFAR-100 classification tasks, DRIFT is compared to PCA, DCT, and raw inputs, showing improved training stability, faster convergence at high learning rates, and smaller generalization gaps. The physics-ML bridge is creative and offers interpretability.

**Strengths:**

1. DRIFT analogizes images to plate vibrations, deriving an orthogonal basis analytically rather than data-driven PCA. This provides interpretability.
2. The comparison experiments demonstrate DRIFT's robustness. It convergence at high LRs where baselines diverge (Fig. 6). On MNIST/CIFAR100, it reduces overfitting with smaller train-val gaps.
3. By enabling effective training with ~2-13% of original dimensions (e.g., 16/784$\approx$2% for MNIST), DRIFT could reduce compute in resource-constrained settings. Preliminary CIFAR-100 results hint at scalability to color images.

**Weaknesses:**

1. The method is evaluated only on low-resolution datasets (MNIST and CIFAR-100) and lacks testing on high-dimensional, noisy, or diverse datasets such as ImageNet or natural images with complex textures and variations, where DRIFT may struggle.
2. Figure 5 shows that at a high compression rate (2%), PCA outperforms DRIFT. At lower compression rates (4–13%), PCA outperforms DRIFT in the early stages of training, and both methods achieve similar performance after 200 epochs.
3. Figure 6 shows that DRIFT outperforms PCA when using large learning rates; however, accuracy drops below 90% on MNIST. This comparison is less meaningful because, in practice, researchers perform hyperparameter tuning. If a model performs poorly, the learning rate would be adjusted.
4. Despite the authors’ claims of efficiency from compact features, they do not report training or inference times, nor provide estimation formulas. This information is critical for real-world applications.

**Questions:**

1. The method was tested only on MNIST and CIFAR-100. Test it on ImageNet or other large-scale image datasets to prove robustness in harder cases.
2. PCA outperforms DRIFT early in training and at low compression rates (2–13%). Does this mean DRIFT slows down convergence?
3. What are the wall-clock training and inference times for DRIFT vs. PCA, and are there closed-form estimates for time/memory as a function of compression rate? The authors should expand the experiments and report the results in a table to compare DRIFT and PCA across compression rates.

| |  compression rates | Training time per epoch (seconds) | Inference latency (ms/sample) | Peak GPU memory (GB) |
|---|---|---|---|---|
|PCA| 2% | -- | -- | -- |
|DRIFT| 2% | -- | -- | -- |
|PCA| 5% | -- | -- | -- |
|DRIFT| 5% | -- | -- | -- |
|PCA| 6% | -- | -- | -- |
|DRIFT| 6% | -- | -- | -- |
|PCA| 13% | -- | -- | -- |
|DRIFT| 13% | -- | -- | -- |

---

> ### Author Response · Authors · 2025-11-27
>
> We do appreciate the time and attention from the reviewer:
> We have updated the manuscript addressing following notes (Q is for questions and W for weaknesses that are not covered in the Questions):
>
> Experimental Scope (Q1):
> -
>
> Added experiments on the CelebA dataset (high-dimensional natural face images) and conducted additional experiments examining the behavior of the method under noisy conditions across all datasets as well as comparison to an AE, see page 8 on the updated manuscript.
>
> Convergence (Q2):
> -
>
> The updated results show DRIFT does not slow down convergence. It exhibits a more stable and less steep training trajectory, reducing oscillations and leading to improved overall generalization and consistently higher final test accuracy within a similar number of epochs considering the times provided on train and test. In the Model section the cost order is provided as well for better clarity on the method cost. We have updated the plots for better and more realistic case scenarios. see page 5,8 on the updated manuscript.
>
> Performance Metrics (Q3):
> -
>
> Added a sample table reporting Training time per epoch (seconds), Inference latency (ms/sample), and Peak GPU memory usage (MB) for all methods it demonstrates the DRFITs stability. Also the order of cost is provided mathematically.
>
> W3:
> -
> We have demonstrated our approach with an emphasis on using a consistent, realistic learning rate rather than comparing multiple adjustable learning rates. Our focus is primarily on evaluating the stability of DRIFT over time, as well as its performance under increased data complexity, including the presence of noise. Noise added don all plots for MNIST,CFIAR, and CelebA.

---

### Meta-Review · Area_Chair_ct2p · 2026-01-21

**Summary:**

The paper doesn't follow the ICLR template. Left and right margins are significantly smaller. This is considered a violation of the ICLR submission policy.

Other than that, the main concerns raised by the reviewers are the significance of the results and the lack of discussion with representation learning methods. To me, while interpreting an image as an elastic plate and using vibrational modes to extract features is interesting, the paper should test their method beyond classification tasks. The reason is that for classification, the "optimal" way to "compress" or do feature extraction is to extract the class labels, which gives the best possible accuracy as well as compression rate.

**Reviewer Concerns:**

The paper doesn't follow the ICLR template even after the rebuttal.

The reviewers were also concerned about the lack of natural image datasets. While the authors provided additional experiments on CelebA, the performance of the proposed method seems to be insignificant.

**Reviewer Scores:**

Reviewer Q2Fw might lower their scores since the authors did not run the requested additional experiments

Reviewers 2my8, BdYW will probably maintain their negative rating.

Reviewer xjep will probably keep the score as most of their comments (e.g., clarification questions) have been resolved by the rebuttal.

---

### Decision · Program_Chairs · 2026-01-26

Reject